# Glacial-Interglacial changes and Holocene variations in Arabian Sea denitrification

Birgit Gaye[1], Anna Böll[1], Joachim Segschneider[2], Nicole Burdanowitz[1], Kay-Christian Emeis[1,3], Venkitasubramani Ramaswamy[4], Niko Lahajnar[1], Andreas Lückge[5] and Tim Rixen[1,6]

[1]Institute for Geology, Universität Hamburg, Bundesstraße 55, 20146 Hamburg, Germany
[2]Institute for Geosciences, Universität Kiel, Ludewig-Meyn-Straße 10, 24118 Kiel, Germany
[3]Institute of Coastal Research, Helmholtz Center Geesthacht, Max-Planck-Straße 1, 21502 Geesthacht, Germany
[4]National Institute of Oceanography, Dona Paula, Goa, 403004, India
[5]Bundesanstalt für Geowissenschaften und Rohstoffe, Stilleweg 2, 30655 Hannover, Germany
[6]Leibniz-Zentrum für Marine Tropenforschung (ZMT) GmbH, Fahrenheitstraße 6, 28359 Bremen, Germany

Abstract
At present the Arabian Sea has a permanent oxygen minimum zone (OMZ) at water depths
between about 100 m and 1200 m. Active denitrification in the upper part of the OMZ is
recorded by enhanced $\delta^{15}N$ values in the sediments. Sediment cores show a $\delta^{15}N$ increase
during the mid- and late Holocene which is contrary to the trend in the other two regions of
water column denitrification in the Eastern Tropical North and South Pacific. We calculated
composite sea surface temperature (SST) and $\delta^{15}N$ ratios in time slices of 1000 years of the
last 25 ka to better understand the reasons for the establishment of the Arabian Sea OMZ and
its response to changes in the Asian monsoon system. Low $\delta^{15}N$ values of 4-7 ‰ during the
last glacial maximum (LGM) and stadials (Younger Dryas and Heinrich Events) suggest that
denitrification was inactive or weak during Pleistocene cold phases while warm interstadials
(IS) had elevated $\delta^{15}N$. Fast changes of upwelling intensities and OMZ ventilation from the
Antarctic were responsible for these strong millennial scale variations during the glacial.
During the entire Holocene $\delta^{15}N$ values >6 ‰ indicate a relatively stable OMZ with enhanced
denitrification. The OMZ develops parallel with the strengthening of the SW monsoon and
monsoonal upwelling after the LGM. Despite the relatively stable climatic conditions of the
Holocene the $\delta^{15}N$ records show regionally different trends in the Arabian Sea. In the western
part of the basin $\delta^{15}N$ are lower during mid- (4.2-8.2 ka BP) compared to late (<4.2 ka BP)
Holocene due to stronger ventilation of the OMZ during the period of most intense southwest
monsoonal upwelling. In contrast, $\delta^{15}N$ in the northern and eastern Arabian Sea rose during
the last 8 ka. The displacement of the core of the OMZ from the region of maximum
productivity in the western Arabian Sea to its present position in the northeast was established
during the mid- and late Holocene. This was probably caused by (i) reduced ventilation due to
a longer residence time of OMZ waters and (ii) augmented by rising oxygen consumption due
to enhanced northeast monsoon driven biological productivity. This concurs with results of
the Kiel Climate Model which show an increase in OMZ volume during the last 9 ka related
to an increasing age of the OMZ water mass.

## 1 Introduction

The marine nitrogen (N) cycle is highly dynamic due to the many chemical compounds of reactive N and their rapid transformation processes (Casciotti, 2016). Its feed-back mechanisms are able to respond to external perturbations, possibly stabilizing the marine inventory of fixed N (Deutsch et al., 2004; Gruber, 2008; Sigman et al., 2009). The range of both oceanic N sources and sinks are still uncertain due to the poor data coverage of rate measurements and the large uncertainties of the water mass ages. The estimates of total N sources and sinks vary by factors of up to four and it has been debated whether the recent marine N cycle is in balance (Brandes and Devol, 2002; Codispoti, 2007; Codispoti et al., 2001; Gruber, 2008; Gruber and Sarmiento, 1997). Models have constrained the major oceanic N sinks (total water column and benthic denitrification) to 120-240 Tg N $a^{-1}$ and brought them close to equilibrium with estimates of diazotrophic dinitrogen ($N_2$-) fixation, the main oceanic N source (Deutsch et al., 2004; DeVries et al., 2013). New measurements have at the same time led to higher global estimates of $N_2$-fixation (Grosskopf et al., 2012).

A period of fundamental change of oceanic N cycling (among other element cycles) occurred during the transition from the last glacial to the Holocene due to adjustment to changes in wind forcing, ocean circulation, sea level, and nutrient as well as trace metal supply from land (Deutsch et al., 2004; Eugster et al., 2013). The present equilibrium was probably attained only a few thousand years ago (Deutsch et al., 2004; Eugster et al., 2013). Understanding the response of the N cycle to this complex reorganization is important to facilitate our present understanding of N cycling on global as well as regional scales (Gruber and Galloway, 2008).

Sedimentary $\delta^{15}N$ values integrate signals derived of N sources and the fractionation processes occurring during N cycling so that $\delta^{15}N$ records have to be carefully deciphered (Altabet, 2006; Brahney et al., 2014; Nagel et al., 2013). Locally, eolian and riverine N supply can impact $\delta^{15}N$ values in sediments (Kendall et al., 2007; Voss et al., 2006) but generally,

sedimentary $\delta^{15}N$ reflect the role of denitrification vs. $N_2$-fixation in ocean basins.
Denitrification in the water column OMZ reduces nitrate in several steps to $N_2$. These
reactions strongly discriminate against the heavy $^{15}N$ isotopes so that the residual nitrate is
isotopically enriched to $\delta^{15}N$ values above the oceanic average of 5 ‰ (Brandes et al., 1998;
Cline and Kaplan, 1975). Convective mixing and especially upwelling force nitrate-deficient
water masses to the surface, so that the enriched $\delta^{15}N$ signal of nitrate is effectively
transported into the euphotic zone. After assimilation into biomass by phytoplankton, $^{15}N$-
enriched particulate matter sinks through the water column to the seafloor where the signal of
denitrification and OMZ intensity is preserved in the sediments (Altabet et al., 1995; Gaye-
Haake et al., 2005; Naqvi et al., 1998; Suthhof et al., 2001). The nitrogen deficit produced by
denitrification can be counteracted by $N_2$-fixation from the atmosphere, which introduces
nitrogen with a $\delta^{15}N$ only slightly lower than the atmospheric value of 0 ‰, as the process is
associated with little isotopic fractionation (Carpenter et al., 1997).
Sedimentary $\delta^{15}N$ records show that during the glacial denitrification was less intense than
today (Galbraith et al., 2013). Models suggest - albeit with many uncertainties and unknowns
- that both denitrification and $N_2$-fixation were lower during the glacial (Deutsch et al., 2004;
Eugster et al., 2013; Galbraith et al., 2013; Schmittner and Somes, 2016; Somes et al., 2017).
However, due to a stronger reduction of denitrification than of $N_2$-fixation, total export
production was higher and increased the glacial oceanic N inventory by 10-50 % over that of
the Holocene, enhancing also the carbon storage in the ocean (Deutsch et al., 2004; Eugster et
al., 2013; Schmittner and Somes, 2016; Somes et al., 2017). Distinct changes of sedimentary
$\delta^{15}N$ values during deglaciation are interpreted to reflect the major changes in the N inventory
(e.g. Galbraith et al., 2013). The decreasing iron supply after the LGM (19-26.5 ka BP; Clark
et al., 2009) may have significantly reduced $N_2$-fixation, leading to a rise of $\delta^{15}N$ (Eugster et
al., 2013). Enhanced upwelling at about 15 ka BP led to abrupt onsets or increases of
denitrification in the eastern tropical north and south Pacific as well as in the Arabian Sea
(Altabet et al., 1995; Ganeshram et al., 2002; Ganeshram et al., 2000; Ganeshram et al., 1995;
Suthhof et al., 2001). The corresponding signal of enhanced $\delta^{15}N$ values was dispersed and
registered in many parts of the global ocean from the glacial to early Holocene and was
followed by a smooth decrease of $\delta^{15}N$ from enhanced $N_2$-fixation stimulated by the delayed
increase of benthic denitrification caused by sea level rise (Deutsch et al., 2004; Galbraith et
al., 2013; Ren et al., 2012). This sequence of events is very well recorded in cores from the
east Pacific upwelling areas, but differs from the temporal pattern seen in sedimentary records
from the Arabian Sea that show stable or increasing $\delta^{15}N$ values in the Holocene (e.g.
Galbraith et al., 2013).
In order to (i) discern why N cycling in the Arabian Sea differs from the global trend and to
(ii) better understand the response of the OMZ to changes in the monsoon system we present
a summary of $\delta^{15}N$-records from the Arabian Sea including two new records from the Oman
upwelling area (Tab. 1; Fig. 1a; supplementary material S1). The records are from different
areas and trace the regional history of mid-water oxygenation over the last 25 ka. To relate the
records of mid water oxygenation to the history of southwest (SW) monsoon upwelling and
northeast (NE) monsoon winter cooling, we compiled SST records from the literature and
generated a new temperature reconstruction for the Oman upwelling area (Tab. 1; Fig. 1b,
supplementary material S2). Based on these integrated $\delta^{15}N$- and SST-records for different
regions of the Arabian Sea we examine contrasts between glacial and Holocene conditions
over the entire basin, and contrasting regional evolution within the basin during the Holocene.
Finally, we discuss our conclusions with results of the global climate and ocean
biogeochemistry model (KCM/PISCES) for the Holocene Arabian Sea.

2 Study Area

The Arabian Sea hosts one of the most pronounced mid-water OMZ of the world's ocean and is a major oceanic N sink due to denitrification and anammox (Bulow et al., 2010; Codispoti et al., 2001; Gaye et al., 2013a; Jensen et al., 2011; Ward et al., 2009). Suboxic conditions between the thermocline and 1200 m are maintained by the balanced interaction of oxygen demand (organic matter degradation) and oxygen supply (ventilation; e.g., Olson et al., 1993; Sarma, 2002). The degradation of organic matter sinking out of the surface mixed layer consumes oxygen in the upper sub-thermocline water column. Primary productivity and particle flux in the Arabian Sea are highly seasonal and more than 50 % of annual particle fluxes occur during the summer season (Haake et al., 1993; Nair et al., 1989; Rixen et al., 1996), when strong SW monsoon winds induce upwelling of cold, nutrient-rich water masses along the coasts of Somalia and Oman (Fig. 2a). Upwelling ceases as changing wind patterns reverse surface circulation from clockwise during the SW monsoon (June-September) to anticlockwise during the NE monsoon (December-March) (Schott et al., 2002). During the NE monsoon the temperature minimum and the productivity maximum occur in the northeastern Arabian Sea off Pakistan (Fig. 2b) caused by deep convection due to winter cooling (Rixen et al., 2005; Wiggert et al., 2005).

Upper water masses (< 1500m water depth) in the Indian Ocean have a net westward circulation while deep waters follow an eastward route as part of the Great Ocean Conveyer Belt connecting the Pacific and Atlantic Ocean (Broecker, 1991). However, this general direction comprises rather complicated routes and pathways (Durgadoo et al., 2017). Four water masses contribute to the subsurface waters of the OMZ: Arabian Sea High-Salinity Water (ASHSW), Persian Gulf Water (PGW), Red Sea Water (RSW), and Indian Ocean Central Water (ICW). The ASHSW forms during the NE-monsoon due to enhanced evaporative cooling driven by dry air from the Himalaya (Prasad and Ikeda, 2002). During September/October the core of the ASHSW at a salinity >36.5 psu and $\sigma_\theta = 23.9$ kg m$^{-3}$ is

found in the upper 100 m in the northern and eastern Arabian Sea (Kumar and Prasad, 1999).
PGW forms a salinity maximum of 35.1-37.9 psu at $\sigma_\theta = 26.6$ kg m$^{-3}$ within the core of the
denitrification zone (100 to 350m) and undergoes strong isopycnal mixing (Prasad et al.,
2001) so that its proportion of the salinity maximum on the 26.6 $\sigma_\theta$ surface is less than 40 %
(Morrison et al., 1998). During the SW-monsoon PGW core salinity is additionally reduced
due to the northward flowing Somali Current providing the less saline ICW (Prasad et al.,
2001; You, 1998). The RSW forms an intermediate salinity maximum between 600 and 900
m and is characterised by salinities between 35.1 and 35.6 psu and $\sigma_\theta$ of 27.2 kg m$^{-3}$ (Kumar
and Prasad, 1999). PGW and RSW are saturated with oxygen from atmospheric contact
shortly before their passage into the Arabian Sea through the Strait of Hormuz (50 m sill
depth) and Strait of Bab-el-Mandeb (137 m sill depth), respectively (Rohling and Zachariasse,
1996; Sarma, 2002). ICW combines Subantarctic Mode Water (SAMW) and Indonesian
Intermediate Water (IIW) and is entering the Arabian Sea from the southwest as part of the
Somali current (Resplandy et al., 2012; You, 1998). Intermediate water from the southern
sources is originally rich in oxygen, but becomes increasingly oxygen depleted and nutrient
rich on its path to the Arabian Sea owing to oxygen loss during the mineralization of sinking
organic matter. Progressive oxygen loss is reflected by the observed pattern with higher
oxygen concentrations in the NW basin than in the NE basin of the Arabian Sea (Morrison et
al., 1999; Pichevin et al., 2007; Rixen et al., 2014). The deep water below about 1500 m is fed
by circumpolar deep water (CDW) with $\sigma_\theta$ of 27.8 kg m$^{-3}$ and a salinity of 34.8 psu (Bindoff
and McDougall, 2000; Schott and McCreary, 2001).
The intensity of the OMZ and denitrification is seasonally variable. Oxygen concentrations in
its core and its volume vary in response to the seasonality of ventilation probably related to
the seasonality of isopycnal mixing (Banse et al., 2014; Rixen et al., 2014). Models produce
similar patterns with major ventilation from the south during the SW monsoon while
reversing currents, progressive oxygen consumption and isopycnal mixing reduce oxygen
concentrations in the entire basin during the winter monsoon (Resplandy et al., 2012; Rixen et
al., 2014). Reoxygenation during the SW monsoon occurs via the invigorated Somali Current
in the western Arabian Sea and a northward flowing undercurrent below 150 m water depth
along the SW coast of India (Resplandy et al., 2012) which was found up to 20°N (Shetye et
al., 1990). This undercurrent reverses its direction semiannually opposing the direction of the
West Indian Coastal Current (WICC; Fig. 2) at the surface (Shetye et al., 1990). At present,
strongest denitrification prevails in the NE Arabian Sea although productivity and particle
fluxes are highest in the western part of the basin (Gaye-Haake et al., 2005; Nair et al., 1989).
Denitrification, that reduces nitrate to nitrite and gaseous dinitrogen, is triggered when oxygen
concentrations fall below 4-5µM $O_2$ (Cline and Richards, 1972; Devol, 1978). In general,
oxygen deficient conditions enable denitrification below 100 m water depth in the Arabian
Sea and active denitrification indicated by the accumulation of nitrite (Naqvi et al., 2008) was
found between about 100-400 m water depth (Gaye et al., 2013a; Martin and Casciotti, 2017).
The intrusion of PGW that flows in a southward direction along the coast of Oman can
occasionally supply oxygen and suppress denitrification (Morrison et al., 1998), as was the
case during the late SW monsoon 2007 between 250 and 400 m water depth (Gaye et al.,
2013a).
Paleoceanographic studies from the Arabian Sea report the existence of a pronounced OMZ
and elevated denitrification during IS and interglacial stages, whereas the Arabian Sea OMZ
was better ventilated and denitrification was suppressed during the LGM and stadials (Altabet
et al., 1995; Higginson et al., 2004; Möbius et al., 2011; Pichevin et al., 2007). Many studies
used productivity proxies and SST reconstructions often in combination with denitrification
proxies such as sedimentary $\delta^{15}N$ values to show that warm periods (IS and interglacials)
were characterized by a stronger SW monsoon inducing upwelling and higher productivity

than cold periods so that denitrification was switched on in the OMZ (Altabet et al., 1999; Altabet et al., 2002; Pichevin et al., 2007; Reichart et al., 1997; Schulte et al., 1999b; Suthhof et al., 2001). After the transition from glacial to interglacial conditions with the warm and cold excursions during the Bølling-Allerød and Younger Dryas (YD; 11.7-12.9 ka BP), respectively, the Holocene was evidently a more stable period of permanent upwelling and denitrification (Böll et al., 2015; Overpeck et al., 1996; Pichevin et al., 2007; Tierney et al., 2016). There are, however, indications that millennial-scale climate oscillations similar to the North Atlantic cold periods detected by Bond et al. (1997), also occurred in the monsoon realm, however, with reduced amplitude (Azharuddin et al., 2017). These Holocene cold periods were found to be characterized by reduced precipitation on land (Menzel et al., 2014) and reduced monsoonal upwelling in the Arabian Sea (Gupta et al., 2003). Volume and intensity of the mid-water OMZ appear to have oscillated related to SW monsoon strength, intensity of winter cooling by the NE monsoon as well as changes in OMZ ventilation (Böll et al., 2015; Das et al., 2017; Pichevin et al., 2007). Thus, understanding Holocene OMZ dynamics is indispensable to evaluate the recently observed OMZ intensification in the Arabian Sea (Banse et al., 2014; Rixen et al., 2014).

Indian Ocean water masses and circulation were quite different at glacial conditions as proxy studies of benthic foraminifera indicate. Deep water was evidently less oxygenated than at present (Duplessy, 1982; Kallel et al., 1988; Schmiedl and Leuschner, 2005; Waelbroeck et al., 2006). This was reproduced by models which showed a generally more sluggish bottom water ventilation from the Antarctic (Rickaby and Elderfield, 2005) with reduced oxygen contents due to the increase in sea ice cover (Buchanan et al., 2016; Somes et al., 2017). A better ventilation of the upper water column in the glacial ocean was explained by the better oxygen solubility in the colder water (Somes et al., 2017). Studies from the southern Arabian Sea furthermore, suggest that there was much stronger formation of AAIW during the

Heinrich Events. This glacial AAIW (GAAIW) would ventilate intermediate waters (~1600
m) in the Arabian Sea where AAIW is not detectable today (Jung et al., 2009).

3 Material and Methods
3.1 Sample collection
The two new core records included in this summary of SST and $\delta^{15}$N records from the
Arabian Sea, are the gravity core SL163 merged with the multicore MC681 taken at the same
location (21°55.97' N, 59°48.15' E, 650 m water depth) and multicore MC680 (22°37.16' N,
59°41.50' E, 789 m water depth). Cores were retrieved in 2007 during Meteor cruise M74/1b
from the continental margin off northern Oman. MC680 and MC681 were sampled in 1 cm
intervals and the first 400 cm of core SL163 were sampled in continuous 3 cm intervals. We
analyzed alkenones in all sediment samples of SL163. $\delta^{15}$N was analyzed in all samples of
MC681 and SL163 and in every second sample of MC680. All sediment samples were freeze-
dried and homogenized prior to chemical treatment and analyses.

3.2 Analyses of the new cores SL163/MC681 and MC680
Stable nitrogen isotopes
The ratio of the two stable isotopes of N ($^{15}$N/$^{14}$N) is expressed as $\delta^{15}$N, which is the per mil
deviation from the N-isotope composition of atmospheric $N_2$ ($\delta^{15}$N = 0 ‰):
$\delta^{15}N = [(R_{Sample}-R_{Standard})/R_{Standard}]*1000$     (1)
where $R_{Sample}$ is the $^{15}$N/$^{14}$N ratio of the sample and $R_{Standard}$ is the $^{15}$N/$^{14}$N ratio of atmospheric
$N_2$. $\delta^{15}$N values were determined using a Finnigan MAT 252 isotope ratio mass spectrometer
after high-temperature flash combustion at 1100°C in a Carlo Erba NA-2500 elemental
analyzer. Pure tank $N_2$ calibrated against the International Atomic Energy Agency reference

standards IAEA-N-1 and IAEA-N-2, which were, in addition to an internal sediment standard,

also used as working standards. Replicate measurements of a reference standard resulted in an

analytical precision better than 0.2 ‰. The mean standard deviation based on duplicate

measurements of samples is 0.07 ‰.

Alkenones

Sample preparation and detailed analytical procedure for alkenone identification are described

in Böll et al. (2014). Purified lipid extracts of between 1.5 to 5 g freeze-dried and

homogenized sediment samples were analyzed for alkenone concentrations using an Agilent

6850 gas chromatograph (GC) equipped with a split-splitless inlet system, a silica column (30

m x 0.25 µm film thickness x 0.32 mm ID; HP-1; Agilent) and a flame ionization detector

(310°C). Alkenone unsaturation ratios were translated into sea surface temperature using the

core top calibration for the Indian Ocean of Sonzogni et al. (1997b):

$$SST = (U_{37}^{K'} - 0.043)/0.033 \text{ with } U_{37}^{K'} = C_{37:2}/(C_{37:2} + C_{37:3}). \tag{2}$$

All lipid extracts were analyzed twice resulting in a mean standard deviation of 0.2°C. The

mean standard deviation of estimated SST based on replicate extraction and measurement of a

working sediment standard is 0.5°C.

Sediment core age models

The age model for SL163 was published by Munz et al. (2017) and is based on 15 accelerator

mass spectrometry (AMS) [14]C datings. In this study the upper 50 cm were taken from

multicore MC681. Four additional datings and visual matching of variations in element

concentrations were used to correlate SL163 and MC681 in their overlapping parts (Tab. S3

and Fig. F1 in supplementary material).

The age model of core MC680 is based on four accelerator mass spectrometry (AMS) [14]C
datings from different core depths, measured at Beta Analytics, Miami/FL (see Tab. S4 and
Fig. F2 of supplementary information). Calibration and reservoir age correction were done in
the same way as for SL163 (Munz et al., 2017). Both cores have a conspicuous sedimentation
hiatus around 5700 years BP. In core SL163 the hiatus was positioned at 57 cm; in MC680
the hiatus was at 37 cm based on a change in facies from olive foraminiferal nanofossil ooze
below to olive brown organic rich nanofossil silty clay above the hiatus.

3.3 Integration and averaging of SST and $\delta^{15}N$ reconstructions
Temperature and $\delta^{15}N$ curves of most cores used here were taken from the literature (Tab. 1)
except those for the new records of cores SL163/MC681 and MC680 presented for the first
time in this paper (see above). All original data and all calculations are presented as Tabs. S1
and S2 in the supplementary material.
In our compilation, temperature reconstructions from the eastern and southeastern Arabian
Sea are based on Mg/Ca ratios of planktonic foraminifera (see methods in e.g. Govil and
Naidu, 2010; Mahesh and Banakar, 2014; Saraswat et al., 2013; Tiwari et al., 2015) except
core MD90963 which has alkenone temperatures (Rostek et al., 1997). All other records are
alkenone temperatures calculated with the core top calibration of Sonzogni et al. (1997b).
Using the published age models, we averaged the temperature records available from the
northern, western, eastern and southeastern Arabian Sea as well as for the Oman and Somali
upwelling systems (Fig. 1b). Composites are based on two to five different core records. The
data were binned in time slices of 1000 years for each individual sediment core. Next, all time
slices of an age interval in a defined study area were averaged. The standard deviations of the
calculated average SST curves rarely exceed the analytical precision of 0.5°C of alkenone
based temperature reconstruction.
Temperature reconstructions based on different methods may differ as proxies may be
seasonally biased or impacted by dissolution or diagenesis (Huguet et al., 2006; Regenberg et
al., 2014). $TEX_{86}$ and alkenone based SST reconstructions of cores NIOP905 and SO42-
KL74 were shown to differ in magnitude and phase as $TEX_{86}$ temperature reconstructions
seem to have a SW monsoon bias (Huguet et al., 2006). Munz et al. (2015) showed that winter
temperatures derived from planktic foraminiferal assemblages had stronger amplitudes than
the alkenone based annual average SST reconstructions in SO130-275KL of Böll et al.
(2014). Two of the records used in this study have both, alkenone and Mg/Ca temperature. In
core P178-15P from the Gulf of Aden alkenone and Mg/Ca temperatures have uniform trends
and are significantly correlated ($P < 0.05$) with a slope of 1 and Mg/Ca are, on average, 0.5°C
lower than alkenone temperatures (Tierney et al., 2016). A comparison of alkenone
temperatures of Huguet et al. 82006) and Mg/Ca temperatures of Anand et al. (2008) of core
NIOP905 also shows that Mg/Ca temperatures are lower. In contrast, the alkenone
temperatures of core MD90963 (Rostek et al., 1997) are about 1°C lower than the two Mg/Ca
temperature records of near-by cores SK129-CR04 (Mahesh and Banakar, 2014) and SK157-
4 (Saraswat et al., 2005) in the southeastern AS. As both, Mg/Ca and alkenone based
temperature reconstructions are calibrated with annual average surface layer temperatures
(Regenberg et al., 2014; Sonzogni et al., 1997a; Sonzogni et al., 1997b) and as we can
identify no trend in our comparison of results of the two methods in Arabian Sea sediment
cores, we have compiled SST reconstructions of both methods.
The average $\delta^{15}N$ values were calculated per time slice in a similar way as SST curves and
averaged for the same areas (Fig. 1a). Before averaging the results of all curves of the selected
areas, $\delta^{15}N$ values were normalized to the average $\delta^{15}N$ value of the respective core (Tab. S1
of supplementary material). Some records were too short to use their average $\delta^{15}N$ values as
they did not cover the main $\delta^{15}N$ shift from the glacial to Holocene. In these cases the
normalization was done with the average value of a near-by core with $\delta^{15}$N in the same range.
This procedure was carried out as $\delta^{15}$N in sediments are impacted by several factors in
addition to the $\delta^{15}$N of nitrate upwelled or mixed from subsurface waters. Nitrogen fixation or
allochthonous supply of nitrogen from rivers or the atmosphere can reduce $\delta^{15}$N in particulate
matter (Agnihotri et al., 2011; Agnihotri et al., 2008; Agnihotri et al., 2009; Lückge et al.,
2012; Montoya and Voss, 2006). Upwelling from different water depths as well as incomplete
utilization of nitrate supplied by upwelling may, furthermore, lead to a gradient with
increasing $\delta^{15}$N values offshore of the upwelling areas (Naqvi et al., 2003). Diagenesis
increases $\delta^{15}$N values in the deep Arabian Sea by up to 3 ‰ in distal sediments (Gaye-Haake
et al., 2005; Möbius et al., 2011). The normalization procedure makes the relative changes in
$\delta^{15}$N comparable within each area despite differences in the diagenetic imprint and in $\delta^{15}$N
sources so that relative changes may be interpreted with respect to the relative intensity of
denitrification. Average $\delta^{15}$N curves of normalized values have a standard deviation of up to
1.5 ‰ with most values far below 1 ‰. The standard deviation is, generally, largest during
deglaciation when $\delta^{15}$N changed rapidly. The curves represent averages of four to seven
individual records except for the Somali upwelling system where only two records were
found. For the construction of the present $\delta^{15}$N chart results from surface samples published
by  Gaye-Haake et al., 2005 were included (Fig. 1a).

3.4 Climate and biogeochemistry model
We use results from an experiment with a global coupled atmosphere-ocean-sea ice model
(the Kiel Climate Model, KCM, Park and Latif, 2009; Park et al., 2008) consisting of
ECHAM5  (Röckner et al., 2003) and NEMO (Madec et al., 2008), to force a global model of
the marine biogeochemistry (PISCES, Aumont et al., 2003) in off-line mode. KCM has been
used for time-slice simulations of the preindustrial and the mid-Holocene climate (Schneider
et al., 2010, Khon et al., 2010, 2012, Salau et al. 2012, Jin et al., 2014). Here, ten times
accelerated orbital parameters (eccentricity, obliquity, and precession) were varied transiently
as forcing according to the equations of Berger (1978). The greenhouse gas concentrations
follow the standard Paleo Modelling Intercomparison Project Phase III (PMIP3) protocol
(Braconnot et al., 2012) based on Indermühle et al. (1999). Changes in the ice sheets are
neglected.
The ocean component (OPA9) uses a tripolar grid with $2^o$ zonal resolution, and a meridional
resolution varying from $0.5^o$ at the equator to $2^o$ x cos (latitude) polewards of $20^o$. The water
column is divided into 31 layers, with 20 layers in the upper 500 m (known as ORCA2
configuration). ECHAM5, the atmospheric component of KCM, is run in T31 resolution with
19 layers, corresponding to a grid cell size of about 400 x 400 km. PISCES (Aumont et al.,
2003) simulates the marine biogeochemistry including processes that determine dissolved
oxygen concentrations based on the oceanic circulation as provided by NEMO (Madec et al.,
2008) and a NPZD-type (Nutrient Phytoplankton Zooplankton Detritus) description of the
marine ecosystem. NEMO/PISCES in the ORCA2 configurations has been used to study
monsoon/biological production interconnections in a recent study by Le Mézo et al. (2017).
Here we restrict the description of PISCES to the processes relevant for the oxygen
concentration. Sources of oxygen are gas exchange with the atmosphere at the surface, and
biological production in the euphotic zone The production of two phytoplankton groups
(representing nanophytoplankton and diatoms) is simulated based on temperature, the
availability of light and the nutrients P, N (both as nitrate and ammonium), Si (for diatoms),
and the micronutrient Fe. There are three non-living components of organic carbon in
PISCES: semi-labile dissolved organic carbon (DOC), as well as large and small particulate
organic carbon (POC), which are fueled by mortality, aggregation, fecal pellet production and
grazing. In deviation to the standard PISCES setup, small POC sinks to the sea floor with a
constant settling velocity of 2 m d$^{-1}$ while large POC settling is simulated depending on the
calcite and opal ballast effect following Gehlen et al. (2006). Oxygen loss occurs through
respiration of organic matter in the entire water column. The respiration rate depends on
temperature with a $Q_{10}$ of 1.8 and on the oxygen level, with a reduced rate for $O_2$-
concentrations below 6 µmol l$^{-1}$. We also use an idealized age tracer that is set to zero in the
surface layer and increases with time elsewhere. Advection and mixing are also applied to the
age tracer. This tracer gives an indication of the subsurface circulation strength. Here we use
it to analyze the change of the average age of the water masses in Arabian Sea OMZ over
time.
We do not attempt to provide a full model analysis of the Arabian Sea OMZ in this paper, but
will mainly use it as an additional tool to estimate the most likely causes for the sediment core
analyses of OMZ intensity changes. Some basic features of the simulated OMZ can be found
in the supplementary material (Figs. F3 to F5 in supplementary material).

4. Results
4.1 Temperature Reconstruction
All temperature reconstructions indicate lower SST during the Pleistocene compared to the
Holocene (Fig. 3). Warming started at about 16-17 ka BP, during the period defined as
Termination 1 (Stern and Lisiecki, 2014) except for the southeastern region where SST rise at
about 22 ka BP in response to rising summer insolation over the northern hemisphere (Berger
and Loutre, 1991). The largest SST increases from the glacial to the Holocene can be
observed in the northern (4°C) and the eastern (3°C) Arabian Sea. The increase is about 2.5°C
in the Oman upwelling area and less than 2°C in the open western Arabian Sea, the Somali
upwelling, and the southeastern Arabian Sea south of India. Some small scale temperature
variabilities exceeding the analytical error of 0.5°C are visible. There is an increase of
different amplitude during the warm IS 2 (~23.4 ka BP) and a small temperature drop during
the YD in the available records of higher resolution (Fig. F6 in supplementary material). In
the average curves (Fig. 3) the YD is visible only in the east and west.
In order to compare the modern and glacial SST distributions (Fig. 4a, b) we plotted the SST
map from the World Ocean Atlas (Fig. 4a; Locarnini et al., 2013) and the time slice at 17-18
ka BP from the core records (Fig. 4b). This time slice is neither an IS nor a Heinrich Event
and has the best data density of the glacial (see supplementary material S2). There is a change
in the SST pattern in the basin between glacial conditions and the Holocene. During the last
glacial, the SST minimum was situated in the northern Arabian Sea as well as Oman
upwelling and there was, generally, a north-south temperature increase (Fig. 3; 4b). During
the Holocene the SST pattern deviates from this north-south increase (Fig. 3; 4a): (i) SST in
the Oman and Somali upwelling areas are lower than northern Arabian Sea temperatures, and
(ii) SST in the eastern and southeastern Arabian Sea are high and in the same range. Small
drops in SST occur in some of the curves at about 9 ka BP and 4-5 ka BP, respectively (Fig.

407  3).


4.2 Patterns of $\delta^{15}N$
The absolute $\delta^{15}N$ values in surface sediments in the present Arabian Sea are elevated with
values between 6 and > 12 ‰ compared with those of the last glacial with values between 3.5
and 7 ‰ (Fig. 5a, b). Holocene $\delta^{15}N$ values are highest in the central part of the basin and in
the Oman upwelling area and lower in most shelf and slope sediments outside upwelling areas
(Fig. 5a and 6). Glacial shelf and slope sediments have $\delta^{15}N$ values below 6 ‰. Similar to the
present situation $\delta^{15}N$ increased towards the center of the basin. However, there are no glacial
$\delta^{15}N$ records from the deepest part of the central Arabian Sea (Fig. 1a; 5b).
The $\delta^{15}$N values increase between 16 and 14 ka BP in all sectors except in the eastern Arabian
Sea, where the increase occurs at about 8 ka BP (Fig. 6). The normalized highest relative
increase of $\delta^{15}$N values by about 3.5 ‰ is observed in the northern Arabian Sea. All other
normalized $\delta^{15}$N curves increase by ≤ 2‰. Most integrated curves show a relative minimum
during the YD when $\delta^{15}$N almost returned to the low glacial values. The general pattern of the
$\delta^{15}$N curves is thus similar to the GISP ice core $\delta^{18}$O record during the glacial and
deglaciations (Fig. 6). The three $\delta^{15}$N curves of high resolution (Fig. F7 in supplementary
material) follow the GISP core with distinct minima during Heinrich Event 1 (H1) and the YD
and a maximum during IS1, whereas IS2 maxima are neither found in the Somali upwelling
area nor in the eastern Arabian Sea (Fig. 6). The Holocene $\delta^{15}$N patterns differ across the
basin. An early Holocene (> 8.2 ka BP; Walker et al., 2012) maximum is observed in the open
western Arabian Sea including the upwelling areas, whereas a late Holocene (< 4,2 ka BP;
Walker et al., 2012) maximum is visible in the northern and eastern part of the Arabian Sea.
An early and late Holocene $\delta^{15}$N peak occurs in the Oman and Somali upwelling areas.

5. Discussion
5.1. Temperature differences between glacial and Holocene
The temperature rise from the LGM to the Holocene in the northern and eastern coastal
regions of the Arabian Sea of  3-4°C is by 1-2°C larger than modelled for the tropical ocean
(Annan and Hargreaves, 2013; Hopcroft and Valdes, 2015; Jansen et al., 2007). This may be
induced by the much lower glacial land temperatures of central Asia (Annan and Hargreaves,
2013) which weakened the SW and strengthened the NE monsoon compared to the Holocene
(Duplessy, 1982). Changes in annual average temperatures in the northern Arabian Sea were
shown to be determined mainly by the intensity of winter cooling and the resulting deeper
thermohaline mixing and thus added to the cooling induced by lower insolation during the
glacial (Böll et al., 2014; Böll et al., 2015; Reichart et al., 2004).
The observed regional differences in temperature rise from the LGM to the Holocene (Fig. 3)
led to a change in the general SST pattern (Fig. 4a, b). The SW monsoon SST pattern in the
modern Arabian Sea with its NW-SE oriented gradient (Fig. 4a) is strongly modulated by
upwelling off Oman and Somalia and inflow of warm and low saline surface water from the
Bay of Bengal via the WICC (Vijith et al., 2016) (Fig. 2a). The WICC inflow is fed by the
North Equatorial Current and starts in the post SW-monsoon period, probably forced by local
winds around the southern tip of India (Suresh et al., 2016). It is related to prevailing sea level
height difference between the Arabian Sea and Bay of Bengal which is due to the enhanced
precipitation and river discharge to the bay (Shankar and Shetye, 2001). A reason for the more
latitudinal gradient of glacial isotherms (Fig. 4b) was a strengthened NE and a weakened SW
monsoon so that winter cooling in the northern Arabian Sea was much stronger (Reichart et
al., 2004) and upwelling was reduced or even inactive during the glacial so that the cold water
source in the western Arabian Sea was strongly reduced (Böll et al., 2015; Duplessy, 1982). In
addition, salinity reconstructions indicate that there was less advection of low salinity, warm
surface waters by the WICC into the eastern Arabian Sea from the Bay of Bengal probably
due to the low glacial precipitation and river run-off (Mahesh and Banakar, 2014).
Glacial SST off Somalia are in a similar range as in the western and eastern Arabian Sea and
by almost 2°C higher than off Oman. This suggests that upwelling off Somalia was weaker
than off Oman or even shut down. At present SW monsoon upwelling is driven by the
positive wind stress curl induced by the Findlater Jet - a low level, cross equatorial jet stream,
recurring during each SW monsoon over eastern Africa and the western Indian Ocean (Brock
et al., 1991; Findlater, 1977). The strength of the Findlater Jet is directly coupled with the
moisture transport to the Indian monsoon region (Fallah et al., 2016). Precipitation on land

such as the All-India Rain Fall is thus used as a measure for SW monsoon strength (Mooley and Parthasarathy, 1984). However, even at present there is no straightforward coupling between high rainfall on land and low SST (Levine and Turner, 2012) and thus also no direct correlation with productivity. Further, sediment proxies indicate and modelling studies suggest that the position of the Findlater Jet changed with monsoon intensity and this could decouple SST, productivity and monsoon strength (Anderson and Prell, 1992; Le Mézo et al., 2017; Sirocko et al., 1991). The Himalayan ice shield during glacial conditions not only led to a reduced temperature gradient between land and sea so that the Findlater Jet weakened, but also to an eastward shift of the continental low pressure cell so that the Jet had a more latitudinal orientation (Le Mézo et al., 2017; Sirocko et al., 1991). Our data suggest that its glacial position was not favorable of upwelling off Somalia.

The SST difference between the northern Arabian Sea on the one hand and the more southern Oman and Somali upwelling areas on the other hand, can be used as upwelling indices as they show deviations from the insolation driven, southward temperature increase (Böll et al., 2015). Enhanced upwelling is indicated by a positive or rising index as it shows lower temperatures in the more southern upwelling areas compared to the northern Arabian Sea. In Fig. 7 we compare it with the index of effective moisture calculated from a large number of lake, peat, loess, and river records from the Asian continent by Herzschuh (2006). Peaks of the upwelling indices at 22-23 ka BP suggest that upwelling prevailed during IS2. During this short warm interval the upwelling was enhanced off Oman and became active for about two millennia off Somalia. The Somali upwelling notably started at about 16 ka BP which could reflect a shift of the Findlater Jet to a position more parallel to the western margin of the Arabian Sea and thus favorable of upwelling in both upwelling areas. The temperature minimum during the YD which is quite pronounced in SST records of high resolution (Böll et al., 2015; Saraswat et al., 2013; Schulte and Muller, 2001; Tierney et al., 2016; see Fig. F6 in

supplementary material) is reflected as a minimum of the upwelling indices at 11-13 ka BP
(Fig. 7). Both, the Oman and Somali upwelling indices increased during further warming after
the YD at 11 ka BP. The moisture index drops after the early Holocene to the present in
parallel with temperature records from the continent (Herzschuh, 2006; Marcott et al., 2013;
Peterse et al., 2014), but upwelling indices remain on about the same level or even increase
during the mid-Holocene (Fig. 7). We surmise that this is a signal of a mid-Holocene shift in
the position of the Findlater Jet. During the warmest period in the early Holocene the
Findlater Jet reached its position closest to the coast so that the Oman upwelling may have
been restricted to a much smaller area (Le Mézo et al., 2017).  In the mid-Holocene the
Findlater Jet shifted offshore and upwelling remained high covering a larger area so that SST
minima prevailed and productivity was enhanced (Le Mézo et al., 2017).
Holocene temperature minima do not coincide in the basin and may have different causes
(Fig. 3). SST minima at near coastal stations in the upwelling centers during the early
Holocene climatic optimum could indicate enhanced upwelling (Böll et al., 2015). The second
Holocene temperature minimum at around 4-5 ka BP in the eastern part of the basin coincides
with a severe draught on the Indian peninsula (Prasad et al., 2014) and colder climate in other
terrestrial climate records from Central Asia (Berkelhammer et al., 2012; Hong et al., 2003;
Ponton et al., 2012). The SST minimum may thus be due to cooling by a strengthened NE
monsoon due to colder winters rather than to warming related enhanced upwelling.

5.2 Nitrogen cycling in the glacial
At present, nitrate reduction between 100-400 m water depths leaves residual nitrate with
$\delta^{15}N$ values up to > 20 ‰ and upwelling can transport enriched nitrate from 250-300 m water
depth into surface waters in the western Arabian Sea upwelling areas (Gaye et al., 2013a;
Gaye et al., 2013b; Yoshinari et al., 1997). Therefore, near shore sediments from the
upwelling area off Oman have $\delta^{15}N$ elevated to > 10 ‰ (Fig. 5a). $\delta^{15}N$ values in all other
recent sediments collected at water depths < 1000 m, i.e. at depths where the diagenetic effect
on sedimentary $\delta^{15}N$ is small or negligible (Altabet and Francois, 1994; Gaye-Haake et al.,
2005), are between 6 and 8 ‰. This is identical to the signal of sub-thermocline nitrate which
feeds productivity primarily via seasonal deep mixing outside the upwelling areas (Gaye et
al., 2013a; Gaye et al., 2013b). The $\delta^{15}N$ values >11 ‰ in the central part of the basin are a
result of (i) offshore advection of $^{15}N$ enriched nitrate from upwelling areas where nitrate is
not completely utilized (Naqvi et al., 2003), as well as (ii) early diagenetic increase of $\delta^{15}N$ in
deep sea sediments (Gaye-Haake et al., 2005; Möbius et al., 2011).
The salient millennial scale oscillation of the Pleistocene $\delta^{15}N$ records call for a strong
mechanisms linking OMZ intensity with northern hemisphere climate oscillations. The low
$\delta^{15}N$ values between 4 and 6 ‰ from < 1000 m water depth of the time slice 17-18 ka BP
(Fig. 5b) and the LGM (Fig. 6; Tab. S1 in supplementary material), suggest that
denitrification was very much reduced or absent. $\delta^{15}N$ values up to 7 ‰ during IS1 and IS2 in
the entire basin except in the eastern Arabian Sea and Somali upwelling area (Tab. S1 and
Fig. F7 in supplementary material) indicate that denitrification was enhanced but restricted to
the northern and northwestern part of the basin during the IS. The $\delta^{15}N$ minima of the YD and
H1 are found in all records of high resolution and the average curves (Fig. 6 and Fig. F7 in
supplementary material). The oscillations of the $\delta^{15}N$ records thus follow the primary
productivity which was enhanced during the warm phases due to stronger upwelling and
reduced during the LGM, YD and Heinrich Events (Leuschner and Sirocko, 2003; Reichart et
al., 1997; Schulte et al., 1999a; Suthhof et al., 2001). Exceptions from this productivity
pattern were reported from the eastern Arabian Sea where the LGM had enhanced
productivity at some locations (Naik et al., 2017), and from the NE monsoon dominated

northern Arabian Sea where productivity was enhanced during the YD and Heinrich Events due to the strong winter cooling (Reichart et al., 2004).

The OMZ sediments from the northern Arabian Sea were indistinctly laminated during most of the last glacial indicating suboxic conditions but were clearly bioturbated indicating fully oxic conditions only during the YD and Heinrich Events (Suthhof et al., 2001). Aragonite preservations and $\delta^{13}C$ of benthic foraminifera further suggest that enhanced formation of GAAIW ventilated the lower OMZ from about 800 m to 1800 m especially during stadials (Böning and Bard, 2009; Jung et al., 2009; Naidu et al., 2014; Schmiedl and Leuschner, 2005). A similar increase of GAAIW formation was observed in the Atlantic and Pacific and was explained with the strong reduction of NADW formation and a simultaneous strengthening of SAMW and GAAIW formation due to reduced salinity and density of North Atlantic surface water (Rickaby and Elderfield, 2005; Ronge et al., 2015). A complete break-down of  the Atlantic meridional overturning circulation (AMOC) during the stadials led to strongest GAAIW production (Rickaby and Elderfield, 2005) and thus to the observed complete OMZ ventilation even in the northern Arabian Sea (Suthhof et al., 2001).  The planktonic nitrate source is at 250-300 m depths in upwelling areas and at even shallower subthermocline depths outside upwelling areas. The water masses feeding nitrate to the surface are thus the ASHSW and ICW. ASHSW formation was possibly stronger when the NE-monsoon was stronger and the climate was more arid so that it better ventilated the OMZ from above. The enhanced formation of SAMW also contributed to better ventilation. At present SAMW is the major oxygen source to the Arabian Sea OMZ while PGW, RSW and IIW are only small contributors of oxygen (Fine et al., 2008; You, 1998). During glacial conditions, the increased SAMW production occurred further north due to the northward shift of the subpolar front similar to the GAAIW (Rickaby and Elderfield, 2005) and better ventilated the upper OMZ (Böning and Bard, 2009). It carried more oxygen due to less

mixing with IIW and RSW and an accelerated circulation (Böning and Bard, 2009) lead to a
lower residence time in the Arabian Sea. Moreover, it is quite possible that glacial SAMW
carried less preformed nutrients due to the better nutrient utilization related to increased eolian
supply of phosphate and trace metals to the region of SAMW formation (Somes et al., 2017).
The lower amount of preformed nutrients further reduced productivity in the Arabian Sea.
Better ventilation and reduced upwelling of nutrient poorer water thus coincided during
stadials and explain the complete oxygenation. The observed suboxic conditions discernible
from laminated sediments in the OMZ during normal glacial conditions (Suthhof et al., 2001)
did not produce enhanced $\delta^{15}N$ signals in the sediments. It is feasible that the oxygen
concentrations did not drop below the threshold of denitrification but it is also possible that
conditions in the Arabian Sea were comparable to those in the present Bay of Bengal, where
nitrate reduction and denitrification occur locally at a low level, but the enriched nitrate is not
transported into surface waters due to stratification (Bristow et al., 2017). Enhanced N
fixation has been suggested as an alternative reason for the low $\delta^{15}N$ found especially during
stadials in the Arabian Sea (Altabet et al., 1995; Emeis et al., 1995; Suthhof et al., 2001). It
could have been stimulated by the supply of excess phosphate and iron from the more arid
continents (Prins, 1999; Sirocko et al., 2000). In this case, N fixation in surface waters
provided N with low $\delta^{15}N$ that may have masked the high $\delta^{15}N$ signal from denitrification.

5.3 Nitrogen cycling in the Holocene
During the Holocene the good coherence with the GISP $\delta^{18}O$ record ceases. The global
oceanic circulation of the Holocene is stabilized by the permanent salinity and density
gradient between NADW and AAIW so that dramatic ocean wide ventilation changes as in
the Pleistocene cannot occur (Keeling and Stephens, 2001). In the Holocene the SAMW
production is reduced so that ICW flowing into the Arabian Sea has a stronger contribution of
IIW (Naidu and Govil, 2010). The considerable $\delta^{15}$N fluctuations by up to 1.5 ‰ indicate that
existing changes of productivity and circulation can still lead to a pronounced Holocene
reorganization of the nitrogen cycle within the basin. The different regional patterns (Fig. 6)
can help to constrain the driving mechanisms. The present pattern of the decoupling of the
productivity and denitrification maximum evolved in the mid- and late Holocene as
denitrification intensified in the northern and eastern part of the basin (Fig. 6b, c). The $\delta^{15}$N
minimum between 9 to 5 ka BP is only found in the western part of the basin. It is most
prominent in the Oman upwelling area and could be a signal of enhanced early and mid-
Holocene OMZ ventilation. Benthic foraminifera indicate that oxygen concentrations were
high and denitrification was low during this period despite enhanced productivity (Das et al.,
2017). The vigorous upwelling during the Holocene climatic optimum was fed by inflow of
ICW from the south which could have better ventilated the western Arabian Sea and thus
compensated for the enhanced respiration (Rixen et al., 2014).
Denitrification has continuously increased during the late Holocene in almost the entire basin
but focused in the northern Arabian Sea (Fig. 6b). This trend coincides with Holocene cooling
and a strengthening of the NE monsoon. Only in the open western Arabian Sea outside direct
upwelling influence, $\delta^{15}$N values decrease in the late Holocene (Fig. 6d). This may be related
to a shift of the Findlater Jet in offshore direction as modelled by Le Mézo et al. (2017) which
may have led to better nutrient availability in the western Arabian Sea. But as there are only
very few late Holocene data from this region (supplementary Tab. S1) this record has to be
interpreted with caution.
Circulation probably changed after the sea level in the Persian Gulf and Red Sea reached its
present position at about 6 ka BP and water masses from these two basins prevented the
strong ingression of ICW to the north-eastern part of the Arabian Sea (Naidu and Govil, 2010;
Pichevin et al., 2007). The ventilation of the OMZ by PGW and RSW today is restricted to

the western part where the OMZ is much weaker than in the northeastern part of the basin (Gaye et al., 2013a; Morrison, 1997). The interplay of reduced OMZ ventilation in the north and enhanced NE monsoon productivity are likely reasons for the relocation of the OMZ and denitrification maximum to the NE during the Holocene. Enhanced productivity in the eastern Arabian Sea since the mid-Holocene as reconstructed from sediment cores (Agnihotri et al., 2003; Kessarkar et al., 2013; Kessarkar et al., 2010) could have added to this relocation. It is likely that the inflow of low density surface water suppressed primary productivity in the eastern Arabian Sea in the early Holocene. After precipitation declined and the sea level difference between the Bay of Bengal and Arabian Sea dropped at about 8 ka BP the inflow of warm low saline water with the northeast monsoon current and WICC to the eastern Arabian Sea declined (Mahesh and Banakar, 2014). This is coinciding with a rise in eastern Arabian Sea $\delta^{15}N$ (Fig. 6c). The OMZ is generally weaker along the west coast of India due to the northward undercurrent which leads to oxygenation of subsurface water during the SW monsoon (Resplandy et al., 2012) and its upwelling and convective transport into surface waters along the coast is likely to explain the low $\delta^{15}N$ in the sediments off the west coast of India (Fig. 5a).

Results from the global climate and ocean biogeochemistry model (KCM/PISCES; section 2.5) driven by astronomical forcing over the Holocene suggest that ventilation changes were important drivers of the late Holocene Arabian Sea OMZ intensification (Fig. 8). The model produces a continuous increase of the OMZ volume in the Arabian Sea from 9 ka BP to the present. This is driven mainly by an increasing age (time since contact with the surface) of the water masses in the Arabian Sea OMZ. Arabian Sea export production is fairly constant in the model (Fig. 8) and can thus be ruled out as the driver for deoxygenation. An increase in export production is modelled only in a small area west of the Southern Indian coast, indicating that export changes may only have played a local role (not shown). The decelerated

circulation allowed more oxygen to be consumed by remineralization, and thus appears to be
the main driver of the progressive deoxygenation in the model (Fig. 8) and can explain the
increasing water column denitrification in the Arabian Sea in the $\delta^{15}N$ records (Fig. 6).
Total organic carbon mass accumulation rates (TOC MAR; Fig. 6g) reflect productivity,
organic matter preservation and burial efficiency (Cowie et al., 2014; Cowie and Hedges,
1993; Müller and Suess, 1979). Similar to $\delta^{15}N$, Arabian Sea TOC MAR deviates from the
global pattern (Cartapanis et al., 2016). Whereas the global TOC MAR significantly declines
during deglaciation and remains low throughout the Holocene, the TOC MAR of the Arabian
Sea shows the decline starting at about 20 ka BP but rises during the mid- and late Holocene
to values similar to those of the glacial (Fig. 6g). This pattern is consistent in the entire
Arabian Sea (see data compiled by Cartapanis et al., 2016). As discussed in detail in
Cartapanis et al. (2016) the drop of TOC MAR during deglaciation may indicate (i) a
reduction of productivity, (ii) a lower transfer and burial efficiency of TOC to the sediments
due to the reduced mineral ballast and the temporary storage on the growing continental
shelves, and (iii) a reduced oxygen exposure time due to faster burial and reduced bottom
water oxygenation of the glacial ocean. As we have discussed above, productivity increased in
large parts of the basin due to a strengthening of the SW monsoon during deglacial warming.
Productivity could thus not explain the reduced TOC MAR after the LGM. A reduced burial
efficiency and increased deep water oxygen content are thus the most likely drivers of the
deglacial TOC MAR drop in the basin. The Holocene rise in TOC MAR, inconsistent with the
global trend, may likewise be due to better preservation caused by progressive mid-water
deoxygenation so that oxygen exposure time again decreased. TOC MAR may have been
augmented by enhanced NE monsoon productivity in the northern part of the basin and
increasing burial efficiency by rising dust supply from the continents due to aridification after
the mid-Holocene (Menzel et al., 2014; Overpeck et al., 1996; Prasad et al., 2014; Sirocko et
al., 1993). The results of the KCM model, however, imply that productivity changes are not
required as the increasing age of the water mass intensifies the Arabian Sea OMZ during the
Holocene.

6. Summary and Conclusions
The compilation of SST reconstructions from the Arabian Sea showed up to 4°C lower glacial
SST compared to the Holocene. Glacial ocean surface circulation in the Arabian Sea was
generally reduced compared to Holocene circulation. Monsoonal upwelling along the western
coasts was very much reduced or absent, as was inflow of low salinity water from the Bay of
Bengal. Therefore, the general temperature gradient had a stronger insolation-driven N-S
trend compared to the circulation-driven NW-SE trend of the Holocene. Upwelling indices
calculated from the temperature difference between the northern Arabian Sea and the Somali
and Oman upwelling centers reveal depleted or even paused upwelling during the LGM and
stadials while upwelling was enhanced during IS. A shift of the Findlater Jet to a stronger E-
W orientation during the glacial could have prevented upwelling off Somalia while it
continued off Oman at a reduced rate. The prevalence of strong upwelling during the mid-
Holocene despite a weakening of the SW monsoon could be due to a shift of the Findlater Jet
in offshore direction.
The compilation of $\delta^{15}$N data shows strong millennial scale oscillations during the glacial
with depleted $\delta^{15}$N values during the LGM and stadials and enriched $\delta^{15}$N during IS. These
oscillations are caused by changes in the OMZ ventilation driven by millennial scale
fluctuations of the oceanic thermohaline circulation. Complete oxygenation of the OMZ
occurred during stadials when SAMW formation was enhanced and it was stronger
oxygenated due to its reduced residence time. The relative instability in thermohaline
circulation led to fast changes in OMZ oxygenation with minima during the IS. In analogy to
conditions in the recent Bay of Bengal, $\delta^{15}N$ of 4-6 ‰ in glacial sediments may not
necessarily indicate that denitrification was completely absent. Moderate or occasional
denitrification may have taken place in a more oxygenated OMZ, but its $\delta^{15}N$ signal was not
recorded in the sediment because the sub-thermocline water mass was isolated from the
euphotic zone by stratification. Also, atmospheric N sources could have contributed to the low
$\delta^{15}N$ of 4-5 ‰ and compensated the enhanced $\delta^{15}N$ in the OMZ.
Holocene $\delta^{15}N$ fluctuations by up to 1.5 ‰ and different patterns in the Arabian Sea show a
strong local reorganization of the nitrogen cycle as global climate and thermohaline
circulation provides more stable conditions. Stronger upwelling in the mid-Holocene was
accompanied by stronger ventilation of the western part of the basin which is also ventilated
by PGW especially after sea level reached its present maximum at 6 ka BP. The present
denitrification maximum in the northeastern part of the basin was formed during the mid- and
late Holocene and is induced by a strengthened NE monsoon due to Holocene cooling but also
due to reduced ventilation of the northern part of the basin. Results of the KCM/PISCES
model simulation show a progressive intensification of the OMZ over the entire model run of
9 ka. Productivity is constant in the model and the main driver of increasing deoxygenation
and denitrification is the prolonged residence time of OMZ waters. OMZ intensification
probably explains increase in TOC MAR throughout the Holocene, also deviating from the
global trend.

Acknowledgements
We are grateful to the German Federal Ministry of Education and Research (BMBF) for
funding the BMBF projects CARIMA and CAHOL as subprojects of the research
programmes CAME and CAME II (BMBF grants 03G0806A, 03G0806B, 03G0806C,
03G0864A). We thank F. Langenberg and S. Beckmann for analytical support. Reiner
Schlitzer and the ODV group are acknowledged for supplying the program Ocean Data View
used for Figures 1, 2, 4 and 5 (Schlitzer, 2016). Computations with KCM were carried out at
the Computing Centre of the University of Kiel. The authors are indebted to the late Ernst
Maier-Reimer for his curiosity and interest in their research over many years, and his never
ending willingness to help and test hypotheses with his numerical models.

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

Table 1: Station number, locations, water depth [m], data sources (references) and
variables used: SST A= alkenone sea surface temperatures, $\delta^{15}$N ratios of total N, SST
Mg/Ca= Mg/Ca sea surface temperatures.

| Core | Latitude | Longitude | Depth [m] | Reference | Variables |
|---|---|---|---|---|---|
| SO130-275KL | 24.8218N | 65.9100E | 782 | Böll et al. 2014 | SST A, $\delta^{15}$N |
| SO90-93KL | 23.5833N | 64.2167E | 1802 | Böll et al. 2015 | SST A |
| SO90-136KL | 23.1223N | 66.4972E | 568 | Schulte and Müller 2001 | SST A |
| M74-SL163/MC681 | 21.9328N | 59.8025E | 650 | this study | SST A, $\delta^{15}$N |
| MD00-2354 | 21.0425N | 61.475166E | 2740 | Böll et al. 2015 | SST A |
| RC27-42 | 16.5N | 59.8E | 2040 | Pourmand et al. 2007 | SST A |
| SK117-GC08 | 15.4833N | 71.0E | 2500 | Banakar et al. 2010 | SST Mg/Ca |
| AAS9-21 | 14.6666N | 72.4833E | 1807 | Govil and Naidu 2010 | SST Mg/Ca |
| SO42-74KL | 14.3210N | 57.3470E | 3212 | Huguet et al. 2006 | SST A, $\delta^{15}$N |
| TY93-929 | 13.1223N | 53.25E | 2490 | Rostek et al. 1997 | SST A |
| MC2-GOA4 | 12.8215N | 46.921666N | 1474 | Isaji et al., 2015 | SST A, $\delta^{15}$N |
| SN-06 | 12.4854N | 74.1265E | 589 | Tiwari et al. 2015 | SST Mg/ca |
| P178-15P | 11.955N | 44.3E | 869 | Tierney et al. 2016 | SST A |
| SK237-CG04 | 10.9775N | 74.999333E | 1245 | Saraswat et al. 2013 | SST Mg/Ca |
| NIOP-905P | 10.76666N | 51.9500E | 1586 | Huguet et al. 2006 | SST A |
| SK129-CR04 | 6.4833N | 75.96667E | 2000 | Mahesh and Banakar 2014 | SST Mg/Ca |
| MD90963 | 5.066666N | 73.8833E | 2450 | Rostek et al. 1997 | SST A |
| MD85674 | 3.18333N | 50.43333E | 4875 | Bard et al. 1997 | SST A |
| SK157-4 | 2.66667N | 78.0E | 3500 | Saraswat et al. 2005 | SST Mg/Ca |
| MD85668 | 0.01667S | 46.0833E | 4020 | Rostek et al. 1997 | SST A |
| MD04-2876 | 24.842833N | 64.008167E | 828 | Pichevin et al. 2007 | $\delta^{15}$N |
| NIOP455 | 23.5506N | 65.95E | 1002 | Reichart et al. 1998 | $\delta^{15}$N |
| SO90-111KL | 23.0766N | 66.4836E | 775 | Suthhof et al. 2001 | $\delta^{15}$N |
| M74-MC680 | 22.6193N | 59.6916E | 789 | this study | $\delta^{15}$N |
| MD04-2879 | 22.5483N | 64.0467E | 920 | Jaeschke et al. 2009 | $\delta^{15}$N |
| NIOP464 | 22.2506N | 63.5836E | 1470 | Reichart et al. 1998 | $\delta^{15}$N |
| NAST | 19.999N | 65.6843E | 3170 | Möbius et al. 2011 | $\delta^{15}$N |
| ODP724C | 18.2833N | 57.4667E | 600 | Möbius et al. 2011 | $\delta^{15}$N |
| RC27-14 | 18.25333N | 57.6550E | 596 | Altabet et al. 2002 | $\delta^{15}$N |
| RC27-23 | 17.993333N | 57.5900E | 820 | Altabet et al. 2002 | $\delta^{15}$N |
| ODP722B | 16.6167N | 59.8E | 2028 | Möbius et al. 2011 | $\delta^{15}$N |
| EAST | 15.5917N | 68.5817E | 3820 | Möbius et al. 2011 | $\delta^{15}$N |
| MD76-131 | 15.53N | 72.5683E | 1230 | Ganeshram et al. 2000 | $\delta^{15}$N |
| SK117-GC08 | 15.4833N | 71.0E | 2500 | Banakar et al. 2005 | $\delta^{15}$N |
| MC2-GOA6 | 14.9800N | 53.767333E | 2416 | Isaji et al., 2015 | $\delta^{15}$N |
| CR-2 | 14.9N | 74E | 45 | Agnihotri et al., 2008 | $\delta^{15}$N |
| SO42-74KL | 14.3210N | 57.3470E | 3212 | Suthhof et al. 2001 | $\delta^{15}$N |
| SS4018G | 13.2133N | 53.2567E | 2830 | Tiwari et al. 2010 | $\delta^{15}$N |

| | | | | | |
|---|---|---|---|---|---|
| SK126-39 | 12.63N | 73.33E | 1940 | Kessarkar et al. 2010 | $\delta^{15}N$ |
| SS3268G5 | 12.5N | 74.2E | 600 | Agnihotri et al., 2003 | $\delta^{15}N$ |
| NIOP-905P | 10.76666N | 51.9500E | 1586 | Ivanochko et al. 2005 | $\delta^{15}N$ |



Figure Caption:
Figure 1: Stations of sediment cores for $\delta^{15}$N (a) and for SST (b) reconstructions with colors
indicating: surface sediment samples (purple); cores from the Oman Upwelling (dark blue);
Somali Upwelling (light blue); the western (green); northern (yellow), eastern (red), and
southeastern (orange) Arabian Sea.
Figure 2: SST in °C from Jan-Mar (NE-monsoon) (a), and Jul-Sep (SW-monsoon) (b) from
the World Ocean Atlas (Locarnini et al., 2013). Solid arrows indicate major wind directions
and broken arrows indicate surface general currents; WICC = West Indian Coastal Current.
Fig. 3: Millennial regional averages of SST [°C] and $\pm1\sigma$ standard deviation in the northern,
eastern, and western Arabian Sea, in the Oman and Somali upwelling areas and the
southeastern Arabian Sea of the last 25 ka. Regions are indicated in Figure 1. Times of high
interstadial 2 (IS2), and low Heinrich Event 1 (H1), Younger Dryas (YD) are indicated by
grey bars. The last glacial maximum (LGM), early Holocene (EH), mid-Holocene (MH), and
late Holocene (LH) are also marked. Lines mark the beginning of MH and LH.
Fig: 4: Annual SST distribution in °C from the World Ocean Atlas (Locarnini et al., 2013) (a),
alkenone and Mg/Ca derived SST reconstruction for the time slice from 17-18 ka BP (b) from
cores shown in Figure 1.
Fig: 5: $\delta^{15}$N in ‰ in recent surface sediments (a); $\delta^{15}$N in sediments at 17-18 ka BP (b) from
surface and core locations shown in Figure 1.
Fig. 6: $\delta^{18}$O  in ‰ from the GISP2 ice core (Grootes and Stuiver, 1997) and sea level [m
above NN] reconstruction from the Red Sea (Siddall et al., 2003) (a), compared with
millennial regional averages of normalized and averaged $\delta^{15}$N [‰] values from the northern
(b), eastern (c), and western (d) Arabian Sea, the Oman (e), and Somali (f) upwelling area,
and total organic carbon mass accumulation in the Arabian Sea (TOC MAR in TgC a$^{-1}$; data
from Cartapanis et al., 2016, and T. Rixen, unpublished) and insolation at 30°N in W m$^{-2}$
(Berger and Loutre, 1991) (g) during the last 25 ka,. Error bars denote $\pm1\sigma$. Grey bars and
abbreviations as in Figure 3; interstadial 1 (IS1).
Fig. 7: Millennial regional averaged SST from the northern Arabian Sea minus Oman
Upwelling averaged SST in °C (black line) and northern Arabian Sea minus Somali
Upwelling averaged SST (red line; regions are indicated in Figure 1.), compared with
reconstructions of the mean effective moisture in southern and central Asia (blue line) and the
Indian monsoon region (green line) from continental archives (Herzschuh, 2006). Grey bars
and abbreviations as in Figure 3.
Figure 8: Simulated volume of the Arabian Sea OMZ (70 µmol l$^{-1}$ threshold, black) and
export production in the entire Arabian Sea (red), western Arabian Sea (green) and eastern
Arabian Sea (blue) over the Holocene (a) and OMZ volume and age of water masses (time
since contact with the surface) averaged over the OMZ (b). Arabian Sea defined as 55$^{o}$E-75$^{o}$
E, 8.5$^{o}$N-22.5$^{o}$ N, border between west and east defined at 68.5$^{o}$ E.

Figure 1:

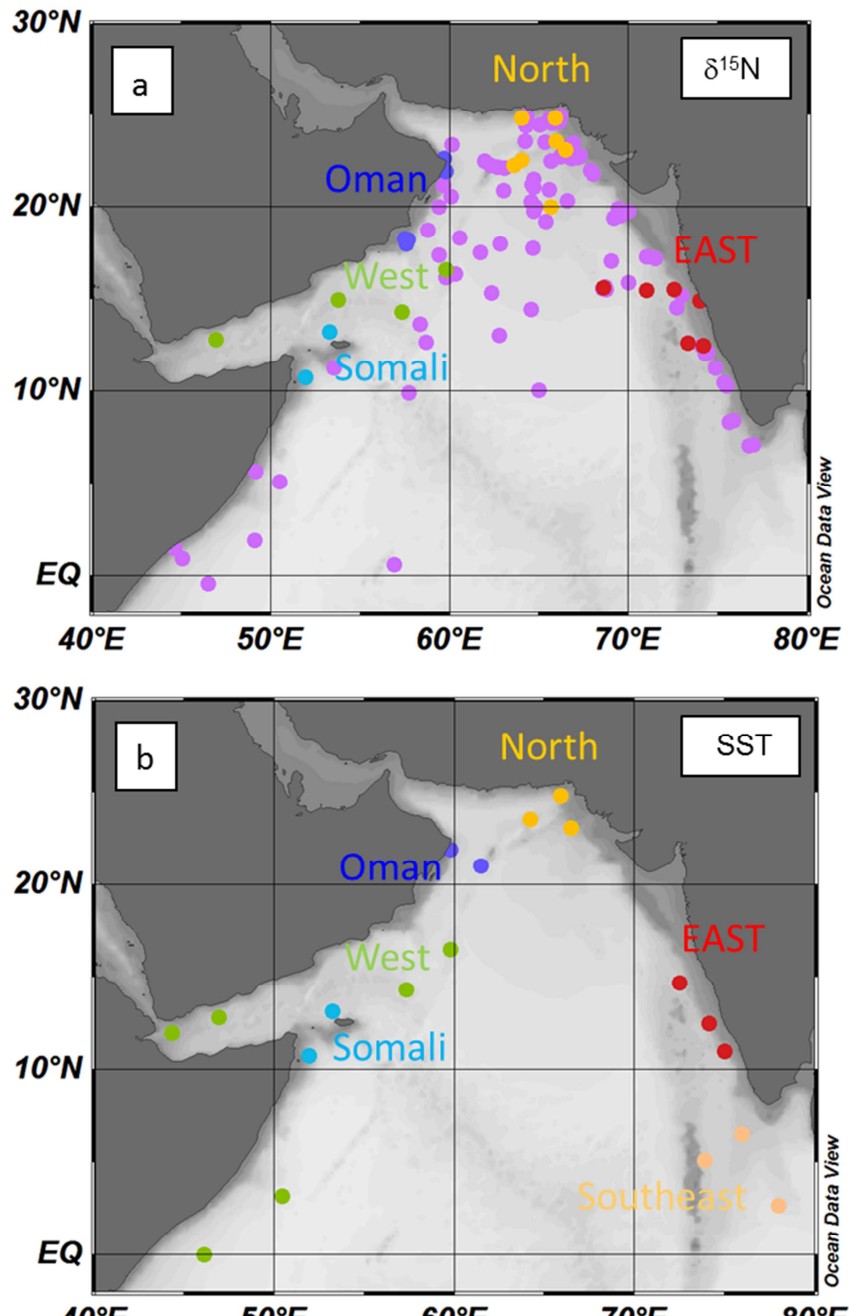


Figure 2:



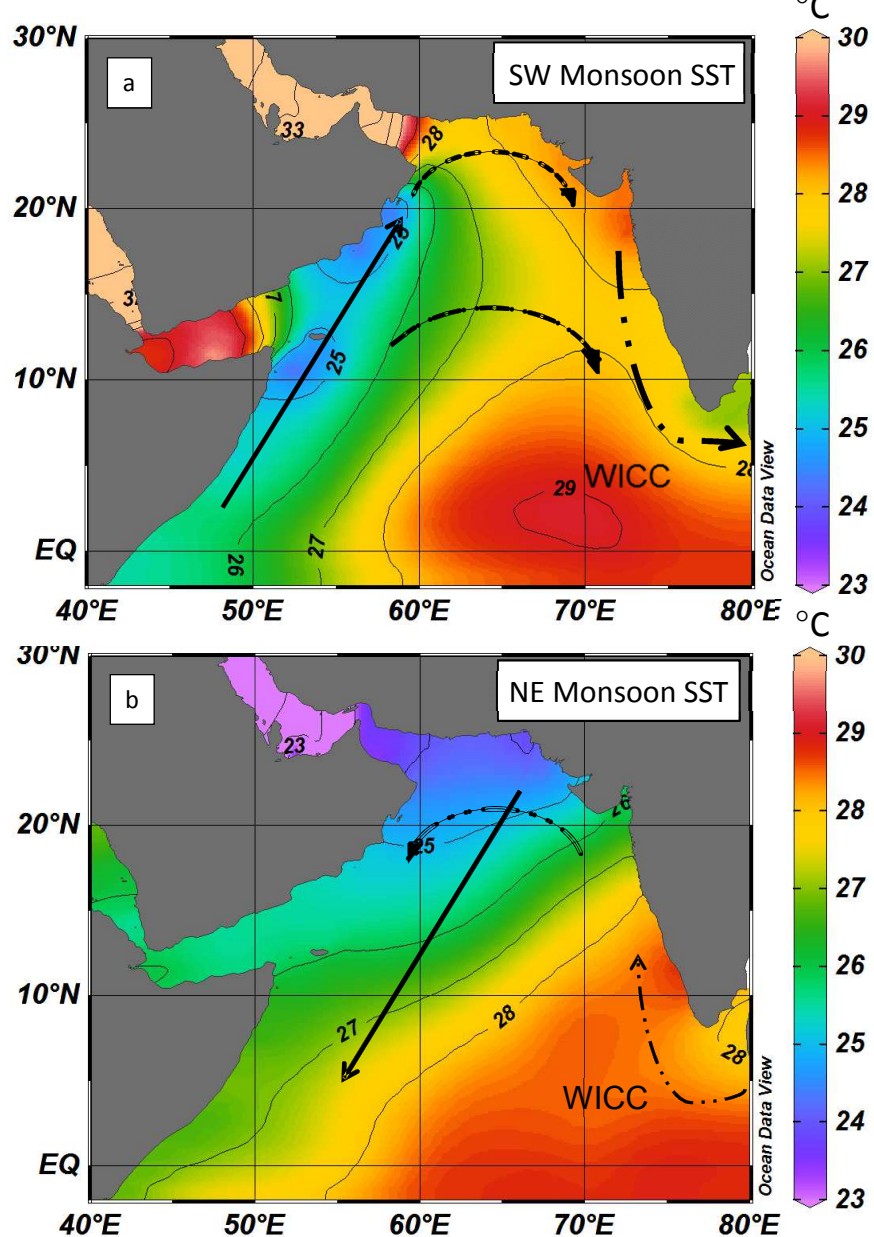

Fig. 3:

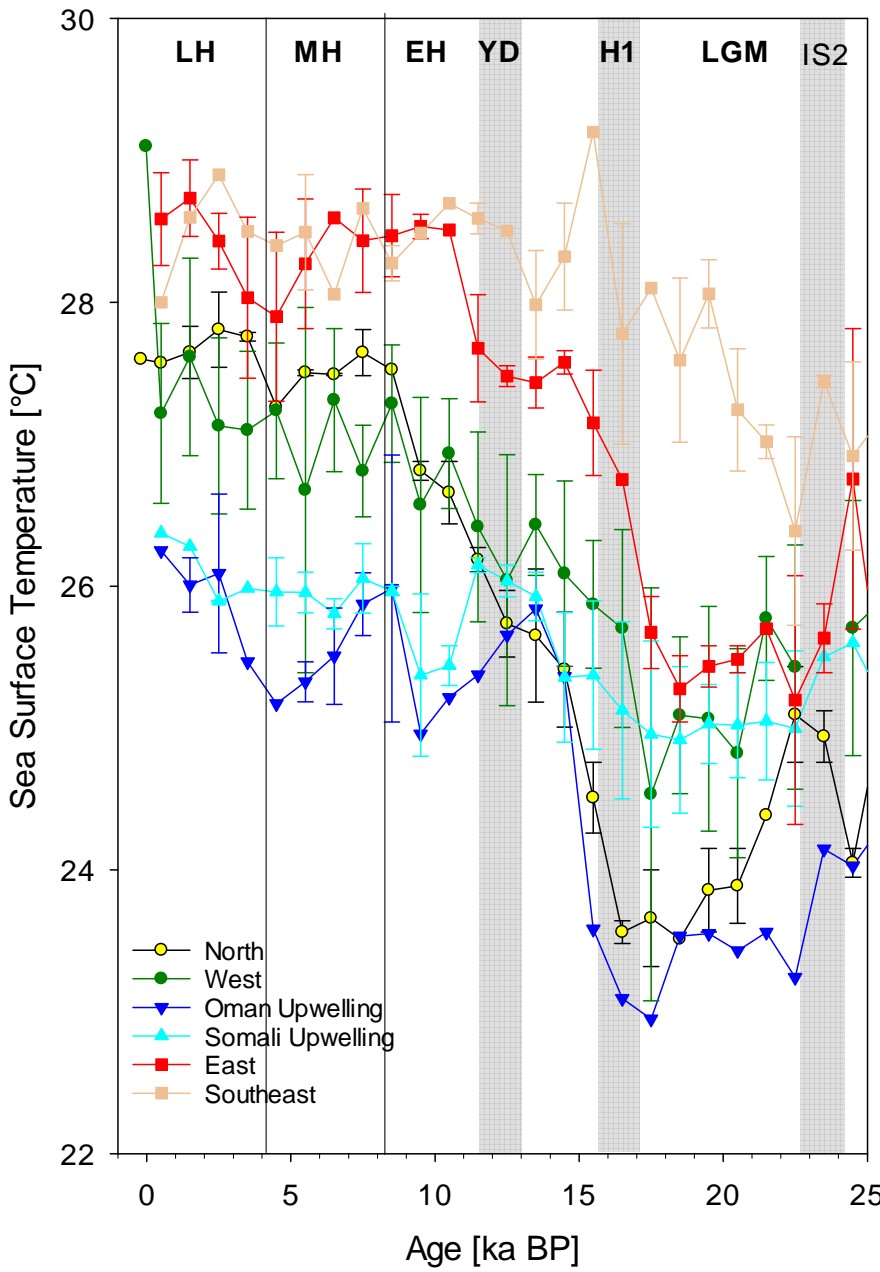




Fig: 4:




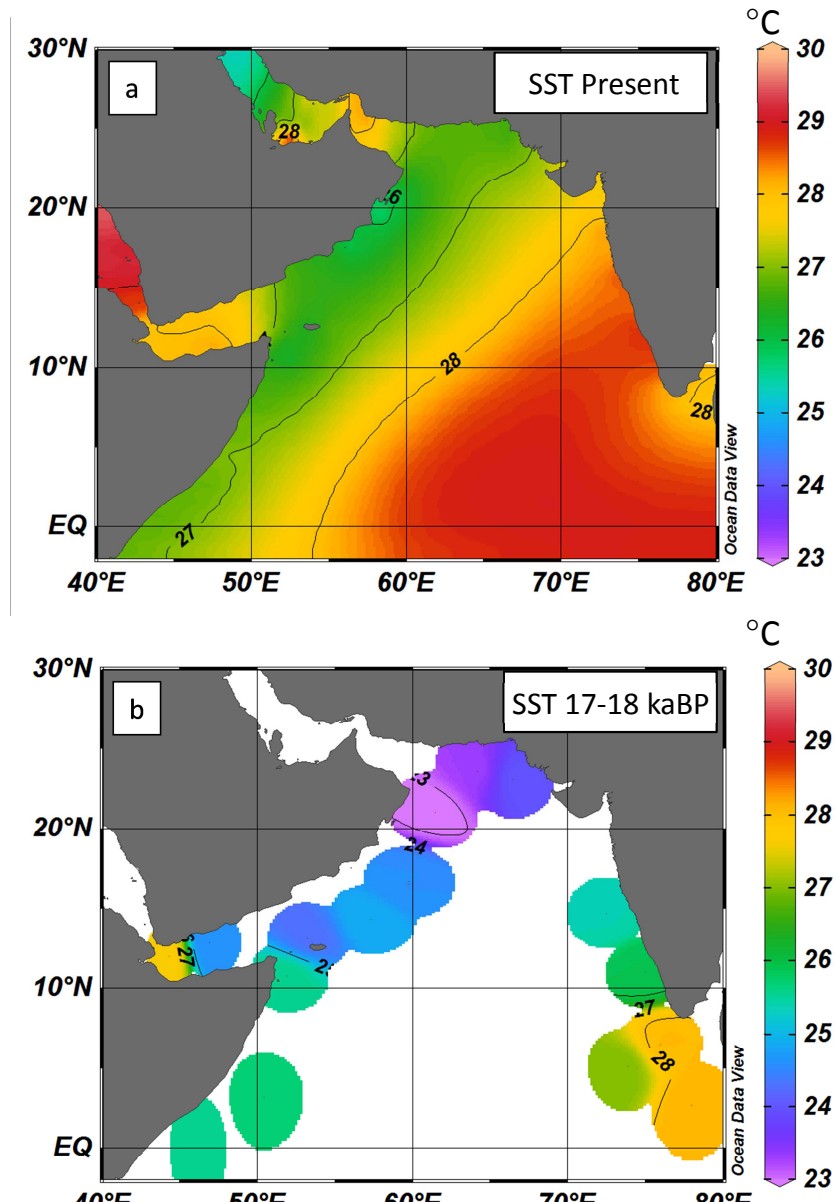

Fig. 5:



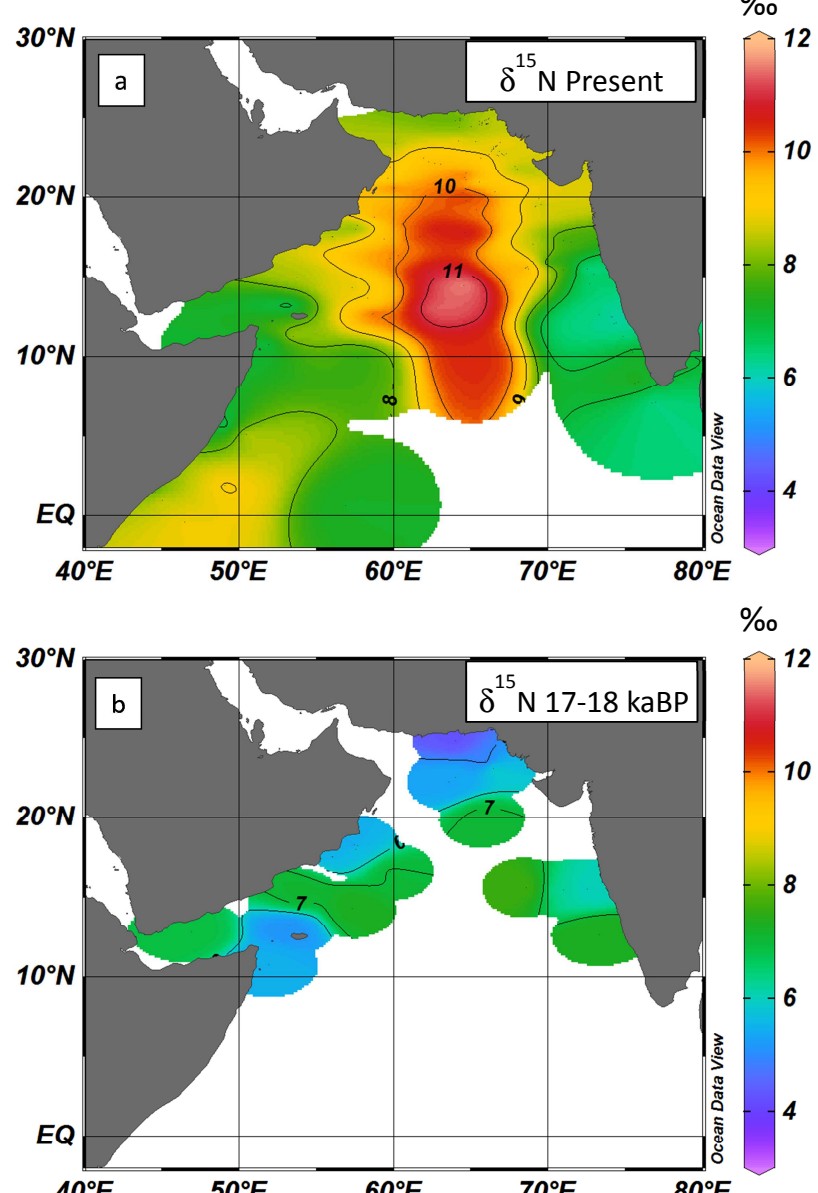

 Fig. 6:

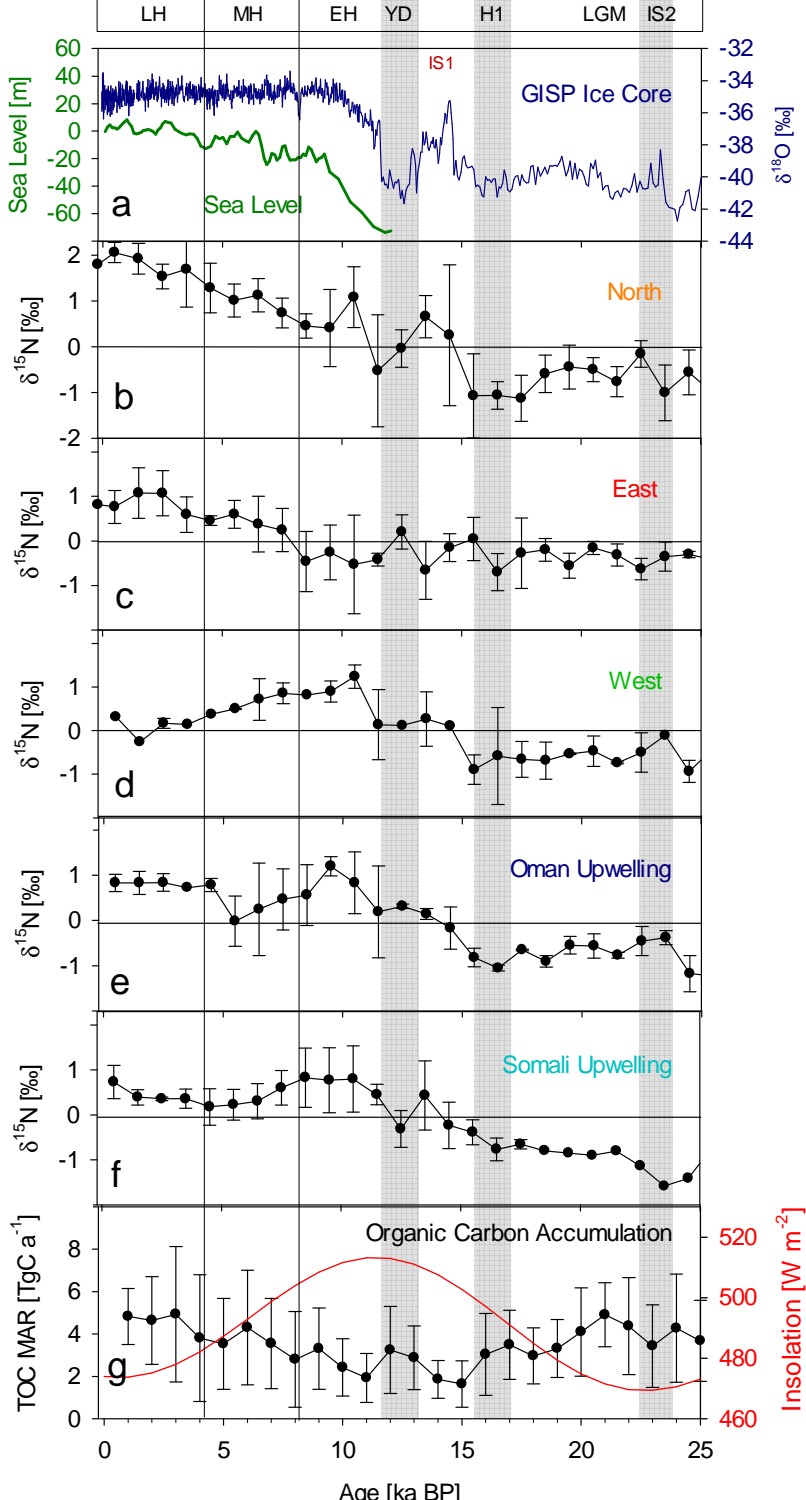

Fig. 7:

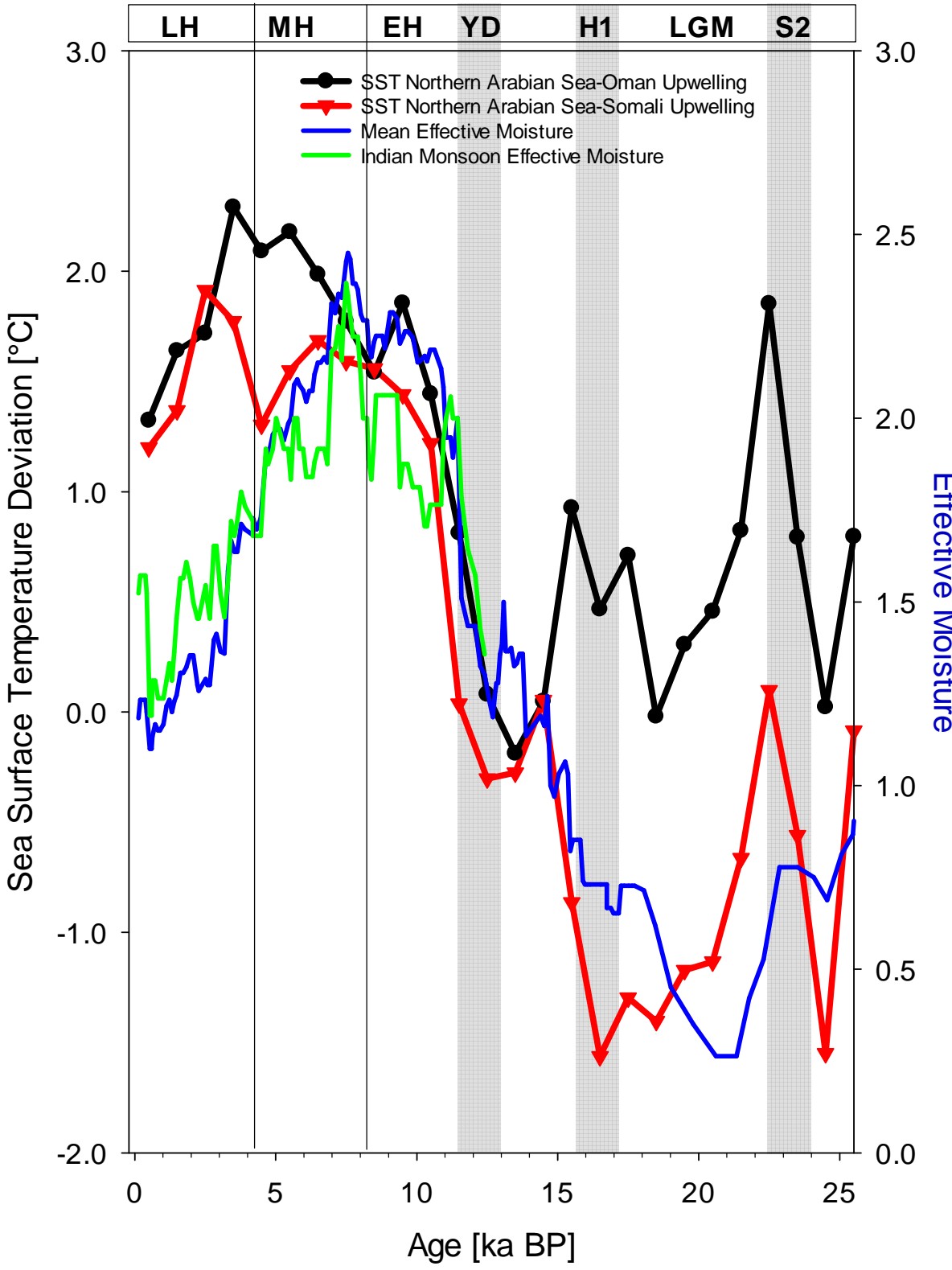

1338

1339

Figure 8:

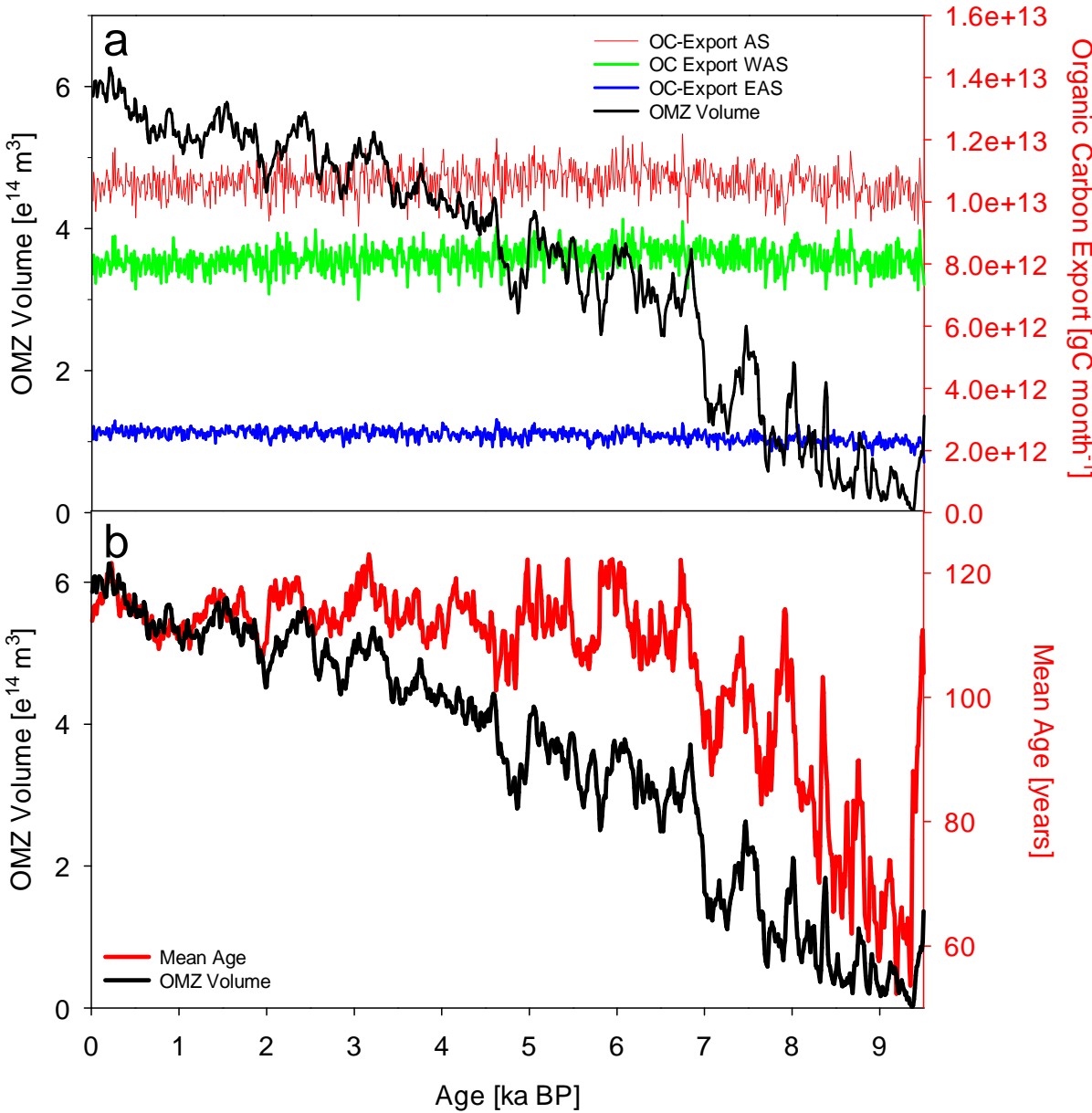