# Peer review of "Glacial-Interglacial changes and Holocene variations in Arabian Sea denitrification"

_Biogeosciences, 2017_

## Referee Comment (RC1) · Anonymous Referee #1 · 8 Aug 2017

**Comments on « Glacial-interglacial changes and Holocene variations in Arabian Sea denitrification » by Gaye et al.**

In this study, the authors analyze the changes in SST and $\delta^{15}$N in two new sediment cores of the Arabian Sea concomitantly with several other cores, during the last glacial interglacial and Holocene periods. They show how the $\delta^{15}$N and SST records reflect the changes in OMZ and monsoon intensities and their interplay.

The study is well conducted, the questions that can arise during the reading are answered in the discussion part and several hypotheses on the changes of $\delta^{15}$N values are discussed. The part analyzing the model simulation lack details and information.

**Specific comments:**

My field of expertise does not extend to the temperature and $\delta^{15}$N reconstructions, so I won't comment much on that part.

Page 11 l239-247, the authors discuss the possible bias due to the differences between the Mg/Ca and alkenone proxies used to reconstruct SST. The authors state that the results from both methods "may thus be comparable" but only one verification has been made.

They argue that a strong correlation has been found between the two methods of reconstruction in one core of the Gulf of Aden. Is it the case for the other cores? How can one good correlation found in one core can justify that both methods are comparable for all the cores used in this study ?

Page 15 l332-335, the authors assume that the lower temperature off Oman compared to the Somali upwelling region during the last Glacial was due to enhanced NE monsoon circulation. Is it possible that a change in the summer wind orientation could also contribute to this pattern in the SST field? If the Findlater jet has a more zonal orientation then it could still sustain the Oman upwelling activity while reducing the Somali upwelling intensity. Some papers such as Sirocko et al. (1991), Anderson and Prell (1992), Bassinot et al. (2011), Le Mézo et al. (2017) have shown that the low-level jet could have moved in the past with potential effects on upwelling and productivity.

Later on page 16 l360-362, can the delay between the moisture decrease and the upwelling index decrease during the Holocene be also somehow explained by a shift in the summer monsoon winds? Can a shift in the wind orientation be able to allow for a sustained upwelling but less moisture transport on land ?

The authors wrote that during the Glacial, the enhanced NEM is supposed to increase convective mixing, which in turn can cause erosion of the OMZ. During the Holocene, the NEM increases from the early to the late Holocene and the result is that productivity increases, which reduces oxygen and intensify the OMZ to the north. Is the difference of increased NEM effects due to the change in circulation in the Arabian Sea between the Glacial and the Late Holocene ?

The authors use a model results at the end of the paper to discuss the residence time of OMZ waters. The reference "Aumont et al. 2003" describes the PISCES model while the reference "Park and Latif 2008" describes the variability of the meridional overturning circulation within the KCM model. I have the feeling that this second reference is not precise enough for the topic of this paper. Is there any paper analyzing the model's ability to reproduce the OMZ in the Arabian Sea ? If not, then a model evaluation is needed here. We need to know if the model is able to reproduce realistic water masses distribution, OMZ extent and productivity/export production distribution in the Arabian Sea to be convinced that the changes it produces throughout the Holocene are not some kind of artefact.

Moreover, how this transient simulation has been realized ? Is there a reference for that?

You can also cite Park et al. 2009, which gives more details about the KCM model than Park and Latif 2008.

References :

Sirocko, F., Sarnthein, M., Lange, H., and Erlenkeuser, H.: Atmo- spheric summer circulation and coastal upwelling in the Ara- bian Sea during the Holocene and the last glaciation, Quaternary Res., 36, 72–93, https://doi.org/10.1016/0033-5894(91)90018-Z, 1991.

Anderson, D. M. and Prell, W. L.: The structure of the southwest monsoon winds over the Arabian Sea during the late Quater- nary: Observations, simulations, and marine geologic evidence, J. Geophys. Res., 97, 15481, https://doi.org/10.1029/92JC01428, 1992.

Bassinot, F. C., Marzin, C., Braconnot, P., Marti, O., Mathien-Blard, E., Lombard, F., and Bopp, L.: Holocene evolution of summer winds and marine productivity in the tropical Indian Ocean in re- sponse to insolation forcing: data-model comparison, Clim. Past, 7, 815–829, https://doi.org/10.5194/cp-7-815-2011, 2011

Le Mézo, P., Beaufort, L., Bopp, L., Braconnot, P. and Kageyama, M.: From monsoon to marine productivity in the Arabian Sea: insights from glacial and interglacial climates, Clim. Past, 13, 759-778, https://doi.org/10.5194/cp-13-759-2017, 2017

Park, W., N. Keenlyside, M. Latif, A. Ströh, R. Redler, E. Roeckner, and G. Madec, 2009: Tropical Pacific Climate and Its Response to Global Warming in the Kiel Climate Model. J. Climate, 22, 71– 92, https://doi.org/10.1175/2008JCLI2261.1

**Minor comments**

Throughout the text there is a mismatch between the Figure 1 labels a and b and the reference to this figure panels: a is referenced as b and inversely (page 5 l 93, l97, page 11 l234, l249)

The name of the new cores varies throughout the text. Be consistent and choose between SL163, 163SL, SL 163, 163 SL also MC680, MC 680

Page 3 l57: you can maybe give a reminder of the definition of $\delta^{15}N$ (shorter than what you do later).

Page 5 l108 : I don't think it is necessary to add the acronym "DNRA" in brackets since you do not use it afterwards.

Page 10 l209: The reference "Sonzogni et al. (1997)" is the reference a or b ?

Page11 l250: "$\delta^{15}$N values were normalized to an average value", what is this average value ?

Page12 l276: "the SST minimum was situated in the northern Arabian Sea", it seems by looking at Figure3 that the minimum is in the Oman upwelling. Do you include the Oman upwelling when you state "the northern Arabian Sea" here?

Page13 l284: "Holocene $\delta^{15}$N values.. (Fig. 5a)" are you describing the values on figure 5a, if so you should use "Present $\delta^{15}$N values" as stated on the figure for clarity. If you are also describing figure 6, then you should state it as "(Figs. 5a and 6)".

Page14 l328: "(Fig. 3b)" I believe you mean Fig. 2b ?

Page 15 l345-350: Do we see a Younger Dryas effect in the moisture index? If not, why ?

Page 15 l354: What do you call "climatic deterioration" ?

Page 16 l361-362: Precise "summer sea surface temperatures … winter SST"

Page 16 l372: "This is identical to the signal of sub-thermocline nitrate, "

Page 17 l386: "time interval"

Page 19 l445-460: Is the TOC MAR record a mean over the whole Arabian Sea ? You should precise where does it come from.

Page 20 l467: At which depth is calculated the export production?
         "(Fig. 8)"

**Comments on the Figures and table**

Table 1:
-   Line of core MD85668 (before the horizontal bold line) : Latitude is -0.01667S you should write 0.01667N to be consistent with the other cores.
-   Line of core NIOP455 (Second line under the bold horizontal line) : $\delta^{15}$N is written d15N in the last column.

Figure 1:
-   You should invert panel a and b to be consistent with the text (or change it throughout the text).
-   You could also add the captions $\delta^{15}$N and SST on the panels.

Figure 2:
-   add the units next to the color scale

- missing reference to World Ocean Atlas in the figure caption

Figure 3 :
- You should keep the same colors as in Figure 1. For the southeast area choose between orange and pink and modify either Figure 1 or Figure 3.
- In the legend, the symbols have a black contour but not the lines in the plot (except for the northern area). You should also be consistent there.

Figure 4:
- Units for the colorbars
- You should invert the panels order since in the text you first discuss panel b and then panel a
- Figure caption : put "(1955-2012)" after "World Ocean Atlas" as in the caption of figure 2. You could add also the reference for this data.

Figure 5:
- Units

Figure 6:
- Missing units for the insolation curve
- Figure caption: you could add the units of the plotted variables as in the other figures' caption

Figure 7:
- Figure caption:
  p33 l904-906 "Millenial regional averaged SST… averaged SST .. averaged SST" or write "Millenial regional SST average … SST average … SST average" or as in the caption of figure 3 "average of SST".

**References in the paper:**
There are cited papers not referenced in the bibliography section, and papers in the bibliography not present in the text.

**IN** the text **NOT** in the biblio :

| Reference | Page and line |
|---|---|
| Altabet et al., 1999 | p7 l159-160 |
| Aumont et al., 2003 | p20 l461-462 |
| Annan and Hargreaves, 2013 | p14 l319-320, p14 l321 (is it a typo for Annan and Hargreaves **2003** ?) |
| Bulow et al., 2010 | p5 l108 |
| Devol 1978 | p7 l147 |
| Haake et al., 1993 | p6 l115 |
| Nair et al., 1989 | p6 l115 |
| Naqvi et al., 2008 | p4 l64 |
| Olson et al., 1993 | p6 l111, p6 l122 |
| Reichart et al., 1997 | p7 l160 |
| Rixen et al., 1996 | p6 l115 |

| Schulte et al., 1999 | p7 l160 |
|---|---|
| Shankar and Shetye, 2001 | p14 l308-309 |
| Ward et al., 2008 | p5 l109 |
| You 1998 | p6 l129 |

**IN** the biblio **NOT** in the text
Annan and Hargreaves, 2003
Budziak 2004
Schlitzer 2016

---

## Referee Comment (RC2) · A. Singh (Referee) · 14 Aug 2017

General Comments Gaye et al., studied changes in the Arabian Sea denitrification on glacial/interglacial and millennial scales and within the Holocene period and demonstrated how the intensity of denitrification coupled with OMZ intensity varied spatially across the Arabian Sea. They also discussed various plausible causes/mechanism (atmospheric and ocean circulations) for temporal and spatial variations in denitrification and the OMZ using sedimentary $\delta$ 15N and SST records. In this effort, authors used their two new sediment cores data from Oman upwelling region and previously published records. Authors suggested that the present OMZ pattern with intensified denitrification in the northeastern region evolved during the late Holocene induced by the intensification in the winter (NE )monsoon coupled with reduced inflow of Bay of

Bengal low-saline water. But, there are certain vital issues remain either unaddressed or less focused while drawing inferences based on various proxies considered in this paper. Specific Comments: Abstract Authors should provide timings (kyr) of events or intervals of major changes, instead of using early, late Holocene, interstadials, stadials etc; as they have data sets with well time constrained. And this approach should be followed through out the manuscript, wherever relevant. 1 Introduction Page3, 51-52; …..during the transition from the last deglaciation to early to mid-Holocene (please mention time interval ?) due to adjustment to changes in ocean circulation and nutrient as well as trace metal supply from land. Variations in Sea level and wind/atmospheric circulation my also have significant influence on nutrient supply ?

2 Material and Methods 2.1 Study Area Page 6, 120-126; please provide data of present day water masses (RSW, PGW) characteristics such as salinity values and O2 concentrations and extent of present day distribution. If possible; add one figure showing surface and deep ocean circulations in the Arabian Sea Page 7, 147—148; In general, oxygen deficient conditions enable denitrification below 100 m (up to what depth ?) in the Arabian Sea (Gaye et al., 2013a). It has been suggested that nitrate reduction occurs between 100 and 500 m depth (Brandes et al., 1998; Ganeshram et al., 2000).

Age Model: (i) Readers would be interested to know the ages of early/ middle, middle/late Holocene boundary. I would prefer to follow concept of Walker et al. (2012) for subdivisions of the Holocene Series/Epoch. I suggest authors should add in the text few lines on this aspect . Also, cite reference for ages of millennial scale climate events used in discussion. (ii) Please elaborate a little bit how a hiatus was identified. 2.4 Integration of averaging of SST and d 15N reconstructions: p.10, 228-229: Authors have missed an important reference (Anand et al. 2008 ;Paleoceanography) and probably this is the first Mg/Ca-SST record from the eastern Arabian Sea. I feel data of Anand et al., 2008, should be included in Table 1, and figures and discussed in the manuscript. 3 Results General: Authors often used words like Glacial; last Glacial, (capital 'G' ?)

; and sometimes IS, IS 2; warm IS 2.....I am really confused with usage of these terminologies and inconsistency throughout the MS . In order to have more clarity; I suggest authors to be precise and mention ages and age intervals and should have consistency throughout the manuscript. P. 13, 289-290: The d15N values increase between.........8ka BP (Fig.6). I think authors are referring to Banakar et al., 2010 eastern Arabian Sea record. Please note that this core comes from much below the OMZ . Please refer core MD 131 (Ivanochko; work) located near lower OMZ. Another important feature of Figure 6 is a close similarity in patterns between northern Arabian Sea record and GISP record: this feature should be incorporated and discussed in the text. 4. Discussion 4.1 P. 13, 301-304: In winter, the WICC ( West Indian Coastal Current) reverses and the Northeast Monsoon Current (NMC) transports waters from the Bay of Bengal (BOB) into the southeastern Arabian Sea up to 130 N (Sarma, 2002; Shankar et al., 2016) only. The high-salinity water (Arabian Sea Water: ASW) is generated in the northern Arabian Sea in winter and spreads southward to the equator with its core at a depth of about 200 m (Kumar and Prasad, 1999; Schott and McCreary, 2001). P.14, 309: Fluvial runoff from western Ghats in summer produces low salinity surface condition in eastern Arabian Sea. This would be another factor for surface water stratification and warming. P.14, 313-317; Are you sure that upwelling was shutdown during the Glacial (I think you mean LGM ?) ; please see Anand et al., 2008. Furthermore , Anand et al 2008 , recorded low-salinity event in the eastern Arabian Sea during the LGM. P.14, 328-329: I do not agree. Anand et al (2008) demonstrated that the patterns and amplitude of SST and SSS changes in off Somali and eastern Arabian Sea varied.

P.15, 343-345: It is again confusing; earlier you stated that upwelling reduced during glacial; Is 22-23 ka event is interglacial or warm interstadial ? I think it is part of the LGM; what could be probable factors for upwelling increase during this short period ? P.15. 351-352; Do you find this temp minimum event in all Arabian Sea records; I think it is not the case in the eastern sector. Please check ? 4.2 P.17, 390-393: The glacial Arabian Sea quickly switched to enhanced denitrification when the SW-
monsoon strengthened . . . . . .. Please be precise with the age of this major switch. Do you mean the warm phases of late glacial or deglaciation ? P.17, 401-406 : Naidu et al (2014, G-cubed) recently suggested intrusion of AAIW (Antarctic Intermediate Water) into the Eastern Arabian Sea and enhanced ventilation during cold Heinrich events. Therefore, this could be an additional factor for a weak OMZ and reduced denitrification. P.18, 407-408: Please explain why there is no increase in Eastern Arabian Sea ; probably due to high fluvial runoff factor ? P. 18, 413-420: Authors record a weak OMZ and enhanced upwelling during 5-9 ka BP and that they related to intensified IOCW inflow. What are plausible factors/hypotheses for increased ventilation during this period. Do authors have some evidence supporting enhanced ventilation through IOCW? P.21, 489-491: In the eastern Arabian Sea (please see unpublished Ph.D thesis of T.S. Ivanochko); denitrification was significantly reduced during cold glacial phases (Heinrich events); not during LGM. General comments: (i) While interpreting d15N records,; authors should also discuss various probable factors that may influence sedimentary N isotope variation such as : incomplete nitrate utilization, Fe limitations; organic matter mineralization in water column and terrestrial input etc. (ii) authors may also consider the hypothesis of changes in Atlantic Ocean overturning circulation driving the nutrient level in subsurface waters (Schmitter et al., 2007).

Minor comments: Please check that references of figures given in the text are correct and properly placed. There should be consistency in naming of sediment cores. Many references cited in the text are not in Reference list and there are references in list not cited in the text. Please check. Page5, 93. . ..history of mid-water oxygenation (what depth interval ?) over the last 25 ka. Page6, 126-129; what is depth of IOCW core in the Arabian Sea ? Page6, 132-133; Progressive oxygen loss . . .. . .. ..Arabian Sea (Ref. ?) Page7, 136-146; I understand that in these lines, authors tried to explain how monsoon wind induced seasonally reversing currents produce spatial changes in oxygen deficient waters and renewal process in the Arabian Sea . I think these sentences to be restructured, so that readers could easily follow. Line 141; . . .a northward flowing undercurrent (please name the current). . .. P.15, 333: What do you

mean by cold phases of glacial ? I think you are referring to Heinrich and stadials events ; Please be specific and mention ages of those events. P.15, 336-338: Authors are contradicting to their previous statement (lines 332-335) where they said that low temp off Oman related to intensification of NE monsoon. P.15. 345-348: I do not understand this sentence ? What are those upwelling indices used by previous workers ? P. 16, 365-368: Please cite original references. P. 18, 410-411 This implies........in the entire basin. But, you stated earlier that eastern Arabian Sea is responding differently? Table 1 Include SK 17 (eastern Arabian Sea ) and 905 (Somali) Mg/Ca SST data from Anand et al., (2008, Paleoceanography) in Table 1. Figures If possible, add surface and deep ocean circulation in Figure 2 or show separately as a new figure. Fig. 3: YD, H1, LGM should be shown with color bands. Holocene subdivisions (early, middle, late) should be marked . Figures 4 and 5 have two panels representing present scenario and 17-18 ka BP. The time-slice 17-18ka BP represents onset of deglaciation after the LGM; when a rapid, major shift occurs. I think authors should briefly state this feature in caption. This would help authors in justifying why they chose this time slice to compare with present day.

Please also note the supplement to this comment:
https://www.biogeosciences-discuss.net/bg-2017-256/bg-2017-256-RC2-supplement.pdf

———————————————————

---

## Referee Comment (RC3) · A. Singh (Referee) · 14 Aug 2017

The comment was uploaded in the form of a supplement: https://www.biogeosciences-discuss.net/bg-2017-256/bg-2017-256-RC3-supplement.pdf

---

## Referee Comment (RC4) · Anonymous Referee #3 · 16 Aug 2017

Comments on manuscript by Gaye et al titled "**Glacial-Interglacial changes and Holocene variations in Arabian Sea denitrification**"

General comments

Gaye et al., have compiled available published data along with two new records from the Arabian Sea to demonstrate changes in the denitrification (using d15N) and temperature (using M/Ca and Alkenones) for the past 25 ka. They have used two new cores (though supplementary data shows three) that has data for ~ 1 Ka in top section, data missing from 1 to 5.6 ka and extrapolated up to 6.1 ka without any dates. The second core also has no record between 2.5 to 5.7 ka with dataset going up to 8 ka. Gaye et al have explained glacial inter glacial variations that is up to 25 ka. For the discussion and interpretation of the data they have combined the their data with other available data in 1 ka intervals but their data and most of the data they have used do not have age model tie points that can give resolution of 1 ka. There are also shortfalls while normalizing the d15N data with average taken for records ranging from 68 years to 25 ka (please see detailed comments age model). I find that most of the interpretation of data is already given by respective published papers. Moreover the temperature and denitrification signals are subdued due to combining dataset region wise (with limitations of dates to give 1 ka accuracy). I would suggest to authors to restrict to Holocene where they have new data (with some additional dates within the hiatus and lower part of the core)

Specific Comments.

**Abstract**

Please modify to give clarity to readers.

e.g. Ln 19-20 "Sediment cores show..... (from which area)...... in other regions (which other regions).

I would suggest to give general trend rather than compiling the data set in 1 ka intervals and restrict to Holocene. Derive interpretation based on your record (so that there will be something new in the data) and then compare with other records.

**Introduction**

Ln37-38 Add reference

Ln39 respond to external perturbation (like what?)

Ln 41 rate measurements?

Ln 42 Water mass age or $O_2$ consumption?

Ln 57-59 Only in certain areas of the world ocean.

Please also add processes like incomplete utilization due to advection, digenetic effect etc..

Ln. 70  Please add "e.g." before Glabraith et al., 2013, as there are many papers stating low denitrification in the Arabian Sea

Ln 77 Please add "e.g." before Glabraith et al., 2013

Ln 78-80 Not clear, How decrease in iron supply at the end of LGM can increase d15N during LGM [but in Ln 69-76 you discuss decrease in d15N]

Ln 83-87- This is a different process compared to open ocean denitrification on which your manuscript is based. It is bit confusing here- if you want to mention about it elaborate a bit so that readers can understand.

Ln 89 Please add "e.g." before Glabraith et al., 2013

Ln-87-89 Is it because of the shallower depths where OMZ is located in Pacific compared to Arabian Sea?

Ln94- Please mention age range (see if you could restrict to Holocene)

Ln 117- Somali, Oman and SW coast of India

Ln 129-130- do you mean IOCW

Ln 133- please add reference (present day or Paleo ?)

Ln 134- The intensity of the OMZ and denitrification (add Ref.)

Ln 142- Add Naqvi et al, 1990 DSR, 37, p593-611; Shetye et al., 1990 J. Mar Res, vol 48 p 359-378

Ln141-145 Naqvi et al., says central Arabian Sea has high denitrification (present day) please correct

Ln159-160 Please see Banakar et al., 2005 high productivity during LGM (Since you are discussing whole Arabian Sea, please discuss this if you take data up to 25 ka--- Please correct the data d15N to Banakar et al., 2005 (Banakar et al., 2010 is wrongly quoted in the supplement)

Ln. 162 Add some Ref. From the eastern Arabian Sea like Naik et al., 2015 GRL, 42, p 1450-1458, Rao et al., 2008, P3, vol 270, p 347

Ln 166  May be in the east, but Naik et al, 2014, Holocene, p 749-755  suggest increasing denitrification

Ln164-166 – Boll et al during Holocene Stable OMZ and Ln 172-174- Ball et al during Holocene changes in the OMZ intensity – Plz correct

**Sample Collection**

Ln 183- Sampled -do you mean subsample of the core or samples analyzed?

Please add total nitrogen data.

Ln 215 Please add more AMS dates if you want to discuss at 1 ka interval changes

**Age Model**

Some part of the data is missing in the supplement please explain how you came to the point of hiatus, it could be slumping of the part of the sediment as your core is located on the slope region.

Please also check models of the other cores used. E.g. Banakar et al., 2005 (you wrongly quoted as 2010) do not have single age

By comparing the Age models of the respective cores you may consider larger time slices

Did you digitize the data? Please mention or source from where you have got the data.

Ln 255-256- Though you have taken four to seven individual curves all data does not cover 25 ka.

Ln 234-235 When you binned time slices please check age models and dates of that particular core

There is another problem with normalization of d15N (Fig. 6)

1) record is for past 68 years – Normalized with d15N average of 68 years

2) record starts at 6 ka ends at 25 ka – Normalized with Average

3) record with missing data 1ka to 5.69 ka – Normalized with Average

4) record with 16 to 24 ka- Normalized with Average

5) record with 10 to 23 ka -Normalized with Average

6) record from 0 to 25 ka - Normalized with Average

There are many records with many combinations my point is "how can you use avg. d15N for the each data set that is not representation of avg of 25 ka"

Water column d15N values changed within past 25 ka and combining and taking average will give biased values.

**Results**

Please explain your new data that will be new in results

**Discussion**

Ln 304- Add NMC in the Figure

Ln 310- Fig. 4a you get this pattern because world atlas does not have data from the eastern Arabian Sea. (Please see the data of the World ocean Atlas)

Ln. 314 "Shut down" I think this is too old reference

LGM taken as 18 to 25 ka and Glacial SST is given at 17-18 (its bit confusing) please explain why this particular time 17-18?

Ln. 330- Please also add clockwise circulation and upwelling by ekman pumping

Ln333- remove inactive

Ln 336-338- There is upwelling in the northern Arabian Sea off Pakistan?

Ln 334- IS2 expand

Ln356-360- not very clear to me

Ln360-362 Why Oman and Somali upwelling are different during B-A event

With limitations in ages (dates) it will be premature to comment on delayed response of the ocean

Ln-378 – Glacial (17-18 ka ) is it not end of glacial?

Ln 408- eastern Arabian Sea- it is because of the data, see kessarkar et al., 2013 clearly shows d15N variations during B-A event.

440-441 HIGH PRODUCTIVITY IS DURING THE SW MONSOON AND low saline water from the Bay of Bengal comes during North east monsoon. Please don't combine the two.

Ln 442-444- Undercurrent Ref. Naqvi et al., 1990 and Shetye et al. 1990

About the undercurrent and oxygenation is mentioned by Kessarkar et al., 2010 and 2013 please use the appropriate reference.

Ln 469- Do you mean southwestern Indian Coast ?

Low Organic carbon in the model may be due to lack of data. High Organic carbon is observed from 15°N to 7°N during LGM cannot be local.

**Conclusions**

High productivity is during the sw monsoon –when currents are in clockwise direction AND low saline water from the Bay of Bengal comes during North east monsoon- when currents are in anticlockwise direction. Both cannot take place at the same time!

Conclusions looks more like continuation of discussion, please try to be specific.

Ln491 Why Bay of Bengal here?

**References:**

Some references mentioned in the text and figures are missing.

**Figures**

Fig. 3- too clumsy-

Fig. 4 Is there any specific reason for choosing 17-18 ka please take full LGM

Fig. 6-TOC does it cover entire Arabian Sea?

30°N for which month and reason for choosing this month?

---

## Referee Comment (RC5) · Anonymous Referee #4 · 18 Aug 2017

General Comments

This manuscript by Gaye et al. presents and discusses sea surface temperature (SST) reconstructions and d15N records across the entire Arabian from the Last Glacial to Late Holocene. They use the SST reconstructions to identify physical mechanisms that drive changes in water column denitrification indicated by d15N. They discuss a complex set of biogeochemical and physical processes that affect oxygen minimum zone (OMZ) dynamics and denitrification including the monsoon-driven effects on upwelling and marine production as well as ventilation by different circulation pathways.

I find this paper to be a nice, well-written discussion investigating how the dynamic monsoon system may have changed and how it affected OMZs and denitrification since the Last Glacial. Their interpretation and discussion of the sediment proxies are generally well-reasoned and support their conclusions. However, I think there are aspects of the manuscript that can be significantly improved before publication that I have noted below.

Specific Comments

1. Introduction and 2.1 Study Area Sections: I think there should be at least a couple paragraphs in the Introduction more focused on the Arabian Sea dynamics and previous paleo d15N interpretations there. The general nitrogen cycle is sufficiently introduced, but given the strong focus of this study in the Arabian Sea and monsoon system, some of the important findings in previous literature should be mentioned here.

Most of this necessary Arabian Sea introductory information is located in the Study Area subsection of the Materials and Methods, which is awkward to me as some readers who are not interested in "Materials and Methods" may overlook this important information including acronym definitions. I recommend moving the last two paleo-related paragraphs of the Study Area subsection into the Introduction and renaming the Study Area subsection to indicate the Arabian Sea oceanographic and monsoon dynamics are introduced there so readers are better informed on that subsection.

Lines 378-395: High interstadial (IS) d15N values are often discussed, but they are not evident in any of the figures. Perhaps it's useful to show the full record (before applying the time-slice averaging) from your new cores in a Figure so these high d15N IS events can be shown. I am not sure which time period(s) and regions they occur in the Arabian Sea. Since this is an important point you make in the abstract and conclusions, I think it should be visibly supported by data rather than only referenced.

Sometimes you refer to "northern Arabian" (e.g. line 384) and sometimes more generally only "Arabian" (e.g. line 385). Since the main contribution of this paper is discussing regional differences, please always specify the region.

Lines 390-391: "The glacial Arabian Sea quickly switched to enhanced denitrification"

Does "quickly" refer to the 16-11ka period? The eastern Arabian Sea does not follow this trend which should be noted.

Lines 411-412: "A short return to glacial conditions without denitrification across the basin occurred during the cold excursion of the Younger Dryas"

I don't see the evidence that supports this statement. The East, West, and Oman records show no significant Younger Dryas d15N decrease (Figure 6). The one point that decreases off Somali is very subtle and still higher than glacial values. The only point below 0 in the North has very high error bars. The SST reconstructions (Figure 3) also do not support a return to glacial conditions.

Lines 417-418: "Evidently, the vigorous upwelling during the Holocene climatic optimum was fed by inflow of IOCW from the south..."

It seems that some other external climate forcing outside of Arabian Sea monsoon dynamics must have caused this enhanced ventilation to overcome the intense upwelling and productivity that would normally cause a strong OMZ. This is an important point to make that could be expanded upon as well as some hypothesis for the mechanism(s) that could be responsible for it.

Lines 421-422: "Denitrification has continuously increased during the Holocene in almost the entire basin but focused in the northern Arabian Sea."

This seems to be an over-generalization to me. All of the western regions ("West", "Oman", "Somali") show a small decrease or no significant change from the Early to Late Holocene.

The selection of the 6 "West" cores in Figure 1b is awkward to me. I wonder if there are important differences between the southern two points near the equator, the western two points near the Red Sea, and the two points between the Somali and Oman upwelling sites since they are in different oceanographic regimes. For example, if ventilation from the Red Sea in response to sea level is a dominant forcing then maybe its

major effect will be most evident in the two points near the Red Sea.

Another question: Where are the values from those two most southern points from the "West" in Figure 5b during 17-18kaBP?

Lines 445-460: Total organic carbon mass accumulation rates (TOC MAR)

I am not sure if the TOC MAR is a useful discussion here, especially since you have not separated it into the western, northern, and eastern regions. Since the standard deviation is so high and as you point out it is unclear whether it indicates enhanced production or preservation, I don't think it helps aide the interpretation of the sediment proxies and thus I am not sure the purpose of discussing its uncertainties.

Lines 461-473: Model results I find the discussion on the modeling results incomplete in terms of comparing it with your interpretation of the d15N and SST proxy records. The model predicts no change to export production in all regions – Isn't this incompatible with your interpretation of monsoon-driven changes to upwelling and organic matter production particularly with respect to the western and eastern basins throughout the sedimentary records (i.e. point (i) in the last sentence of the abstract)?

Does the model simulate a realistic Arabian Sea monsoon system and OMZ in the modern ocean? What causes the decelerated circulation in the model: Is it a large-scale effect relating to a slowdown of the high-latitude subduction water masses or changes to more local currents?

Discussion and Conclusions: It is not clear to me what the new findings are compared to all of the conclusions from previous literature that you reference. Does your integrative view comparing all of the Arabian Sea regions reveal new findings/mechanisms/controls on the OMZ and denitrification that has not been discussed before? They should be specifically pointed out and emphasized more throughout the Discussion section and in the Conclusions.

Minor Comments

lines 72-73: Schmittner and Somes, 2016 reference The Schmittner-Somes team has a more recent, realistic model study on the glacial nitrogen cycle that would make a more appropriate citation here than that 2016 paper.

Somes CJ, Schmittner A, Muglia J and Oschlies A (2017) A Three-Dimensional Model of the Marine Nitrogen Cycle during the Last Glacial Maximum Constrained by Sedimentary Isotopes. Front. Mar. Sci. 4:108. doi: 10.3389/fmars.2017.00108

Lines 85-86: ". . . smooth decrease of d15N induced by the delayed increase of benthic denitrification cause by sea level rise. . ."

N2 fixation would be required to decrease d15N since benthic denitrification alone would slightly increase d15N because it slightly fractionates the isotopes. I suggest rephrasing to "smooth decrease of d15N from enhanced N2 fixation stimulated by the delayed increase. . . "

Lines 107-108: "dissimilatory nitrate reduction to ammonium (DNRA)"

DNRA itself is not an oceanic N sink process because ammonium is a readily bioavailable N pool so this process should not be included here.

Line 164: ", respectively" – More readable sentence if moved after "Younger Dryas"

Line 345: "interglacial" – Do you mean "interstadial"?

Line 402: "prograde" – perhaps circulate or propagate is better

Lines 441-444: You mean that there is no OMZ in the SE, right? Then I recommend rephrasing to

"Today, the OMZ is absent in the SE due to a northward undercurrent. . ."

---

## Author Comment (AC1) · 21 Sep 2017

Reply to reviewer 1:

Thank you very much for your detailed comments which will help to improve the ms. Special thanks for checking the errors in Figures, their labels, references and many small mistakes.

Page 11, lines 239-247: P178-15P is the only core for which we found a highly resolved temperature reconstruction by Mg/Ca ratios of foraminifera and from uk′37 (alkenones). Reviewer 2 has also brought to our attention that there are alkenone and Mg/Ca records for core NIOP 905 also. One of the southeastern records (MD90963; Rosteck et al., 1997) is, moreover, an alkenone record and it shows a similar trend as

the near-by Mg/Ca records. This information will be added to a revised version. The high correlation of these records does, however, not mean that all records would be comparable in the other cores and we will delete this statement (page 11, line 245/46).

Page 15, lines 332-335: We agree that we have not given enough credit to the previous work on a shift of the position of the Findlater Jet and think that this could add to the better understanding of the glacial SST pattern as well as to an interpretation of the decoupling of upwelling and moisture transport. The four suggested papers and their finding will be included in the discussion and will help to come to a more convincing explanation for the glacial/interglacial changes and the observed differences between the Oman and Somali upwelling.

As the reviewer suggests, some of the differences of the effect of NEM strength during the glacial and Holocene must be related to different circulation/ventilation patterns. As reviewer 2 has also suggested a section of water mass structure will be added and we will also elaborate on the available information on glacial mid- and deep water ventilation. We will add a model description to the Methods including further references describing the model and its application. We will add two Figures to the supplement which compare the reproduction of the OMZ in the model with WOA data.

Minor comments

Figure labels and core labels will be checked and made consistent.

Short introduction to d15N will be added and the process of averaging will be further described, including the reference to the supplement.

The use of northern Arabian Sea, respectively Oman upwelling will be checked.

A more detailed description of the source of TOC MAR data will be included.

All other minor comments will be dealt with, Tables and Figures will be checked and change of Figures and captions will be done; references will be carefully checked.

---

## Author Comment (AC2) · 21 Sep 2017

Reply to reviewer 2:

Thank you very much for taking the time for your detailed comments which will help us to make the ms. more readable. As you suggest we will add a chapter on water masses in the Arabian Sea which will meet a number of your specific comments.

Abstract

In the abstract and throughout the ms. we will check and add the correct timings of glacial and Holocene events, times etc..

Introduction

[Figure]

We will include variations in sea level and atmospheric circulation as drivers of changes in nutrient inventories.

Materials and Methods

Page 6/7: A paragraph on water masses will be added describing their characteristics. We will also describe the water depth of oxygen deficit and denitrification in more detail. We will add the ages of the Holocene boundaries and a few lines on the times of millennial scale oscillations. The hiatus was identified by a facies shift. This will be described in detail. The results of Anand et al. (2008) will be included and discussed. These data can also be used for a comparison of alkenone and Mg/Ca SST of NIOP 905 (see reply to reviewer 1).

Results

Ages and precise and consistent terminologies will be used. An increase of d15N around 7-9 ka BP can be observed in all cores from the eastern Arabian Sea (including MD 131). As Möbius et al. (2011) showed that there is a diagenetic increase of d15N in deep cores which may change the absolute values but will change the general trend only if diagenesis is extremely variable with time. If this is not the case the trends will still be visible. We intended to minimize such differences in diagenesis by our normalization procedure. Yes, we will point out the similarity with the GISP record.

Discussion

P 13: We will add more details on the exchange of water masses between the Bay of Bengal and Arabian Sea and the seasonal changes. The possible role of the Western Ghats will also be mentioned.

P14/15: We believe that upwelling was shut down during the period we studied (See Böll et al., 2015) because there is hardly any SST difference between the present upwelling areas and the northern AS. We further believe that exceptions are the interstadials defined in the GISP core which were shown to coincide with TOC peaks

(Schulze et al., 1998) and d15N peaks (Altabet et al., 1995, 2002; Möbius et al., 2011) in cores with high resolution. These peaks can be explained with short intervals of upwelling. SST records of high resolution show that IS 2 (∼22-23 ka) was a cold period in upwelling areas in the western Arabian Sea. The eastern AS records to not show this minimum. We will try to make this clearer in the revised ms.

P17 we will indicate the exact times.

P 17/18/21: the discussion of water masses (see above) will also include the possible ventilation by AAIW during Heinrich events and check especially the situation in the eastern Arabian Sea.

A short detailed introduction to the use of d15N as an indicator of N-cycling processes will be added (see comments to reviewer 1). The processes influencing d15N will also be discussed.

Minor comments

References, figures, tables and captions will be rechecked. References will be added were required, more information on water mass structure will be included.

We feel that there is not enough information so that we cannot present a deep circulation of the Arabian Sea but will check in the literature if this is feasible.

Anand et al. 2008 will be included (see above).

---

## Author Comment (AC3) · 21 Sep 2017

Reply to reviewer 3:

Thank you very much for your review and the detailed comments which we will carefully go through in our revised version. We greatly benefit from four the four detailed reviews of colleagues who are very well acquainted with the study area so that we can critically check the entire ms. and avoid mistakes and unprecise statements.

Reviewer 3 questions whether the binning of data into 1 ka intervals is useful and also points out that some of the records have few age tie points while other records are very highly resolved. Due to these differences in resolution we decided to bin the data in 1 ka intervals. The idea was to use this procedure to eliminate some of the local

variations and identify the main trends. But we think that it is a good idea to strengthen our points by showing some of the highly resolved records in the revised version (see also comments to reviewer 4).

Another suggestion of reviewer 3 is that we should look at the Holocene only. We think that looking at the last 25 ka is one of the main focusses of this paper. Our idea was to provide a comprehensive picture of glacial-interglacial changes for the entire basin and the Holocene development and to identify the main drivers of changes in the N cycle. We have received many comments by Dr. A. Singh and the other reviewers to be able to improve this version of the ms. with a better presentation of the role of circulation/ventilation changes, of the reasons for correcting d15N, of the applicability of this tracer and of model results.

Lines 19-20 will be clarified, we will refer to ETNP and ETSP upwelling

Line 37-38 we will add ref.

A description of the N cycle will be added to the intro (see reviewer 1) which will give us the chance to satisfy most of your requests.

Line 79-80 will be rephrased.

References added where required and Banakar et al. 2005 vs. 2010 will be rechecked.

Naik et al., 2014, 2015 will also be checked

Sample collection

Line 183: Data on core SL 163 and MC680 will be made available on Pangaea – data repository.

Line 215: The age model of SL163 was published by Munz et al. 2017 and the hiatus was identified by change of facies. TOC and N curves could be paralleled between the cores so that a relatively good age model becomes available also for MC680 (short multicore).

The d15n data are from Banakar er al. (2005) and the SST curve was from Banakar et al. (2010) made available by Dr. Banakar.

The normalization procedure was carried out in order to be able to look at changes in d15N sources and eliminate the local biases. In the revised version we will discuss the possible biases in more detail. In the supplement we have indicated the average value used for normalization for each core. Averages were in a similar range even when cores did not cover the entire 25 ka but when the main shift in d15N values from the glacial to Holocene conditions was covered. In some cases the records were indeed too short to use the average of the particular core. In this case the average was derived from near-by cores when the d15N values in the available time periods were in the same range. All values used for normalization are shown in the Supplement and we will recheck them in the revised version.

The new data are indicated in Table 1 and the supplement and will be available on Pangaea. They are a small contribution to the d15N records but one of only two SST records available for the Oman upwelling region.

Dates and terminology will be checked.

A paragraph on water mass structure will be added (see above, reply to reviewer 2) and the seasonal circulation off the west coast of India will also be expanded. Kessarkar et al. 2010 and 2013 will also be discussed.

References will be checked (see review 1).

Fig. 4: data availability is best for 17-18 ka and this period has lowest temperatures of the glacial period

TOC covers the entire Arabian Sea (see review No. 4).

Changes suggested in the many other minor comments will be made.

---

## Author Comment (AC4) · 21 Sep 2017

Reply to reviewer 4:

Thank you very much for the review and the valuable suggestions which will help to considerably improve the manuscript.

As reviewers 1 and 2 also suggested we will expand the introduction of Arabian Sea water mass structure and dynamics. In the revision we will check if the presentation of some highly resolved individual records, as has also been suggested by reviewer 3, could strengthen some of our points. We will also add more literature on N-cycle in general and specific Arabian Sea literature. A new organization of the Introduction and study area and the removal of the latter from Materials and Methods to make it more

visible will be another improvement.

Lines 378-395: we will check records and show those with very high resolution (possibly in the Supplement or as a new Figure in the main paper) to make this point (high d15N during IS) more convincing.

Terminology will be checked and corrected throughout the ms.

Lines 390/391: this refers to the IS events during the glacial conditions.

Lines 411-412: we will check this and will either eliminate this or show individual records if they clearly show this.

Lines 417-418: We will expand on this point of remote forcing.

Lines 421-422: we will be more specific. The major change is in the northern Arabian Sea.

The two southern "west" cores have troubled us for some time during ms. preparation and as the SST records are not very different from the more western AS records off Oman we binned them together with the former. But, as Figure 4 implies Holocene SST are very similar at the two southern locations while glacial SST are higher than at the other locations. We will check if there is additional information when plotting these two locations separately. Unfortunately, there are no d15N data from these two locations.

Lines 445-460: We will check if it makes sense to separate the records into west, north etc. or otherwise delete the TOC MAR.

Lines 461-473: A comparison of model results with WOA data will be included and we will work on the model-data comparison and try to evaluate remote forcing vs. local effects on the OMZ.

Discussion and Conclusion

We believe that the contribution of the paper is to provide a synoptic view of the processes responsible for the glacial-interglacial as well as the stadial-interstadial increases in denitrification and the regional variability within the basin. The other and more important contribution is that we can show that the present position of what is often called the core of OMZ and denitrification in the northeastern Arabian Sea is evidently a recent development. We suggest that this is due to a strengthening of the NE monsoon but also and may be more important to remote forcing. We will concentrate on these aspects in the revised version.

Minor comments

Lines 72-73: Sommes et al. 2017 will be cited (and read)

The remaining minor changes will be made.

---

## Author Comment (AC5) · 21 Sep 2017

Dear reviewers,

We thank the four reviewers for their valuable suggestions and the time taken for the reviews. There are four major points evolving from the reviews which we will be able to address in a revised version:

- A chapter/paragraph on Arabian Sea water masses and circulation (surface and deep) has to be added which will also help to improve the discussion of results with respect to local vs. remote forcing.

- The introduction to the N cycle will be enlarged with more details and references to earlier work.

[Figure]

These first two points will lead to a slight reorganization of the initial chapters Introduction and Study area

- More information on the model has to be presented in the methods and a model-data comparison has to be shown and discussed.

- The major trends in AS nitrogen cycling and the new contributions to this have to be emphasized throughout the manuscript (some of the suggested literature will help).

Most of the many minor suggested changes, such as adding dates in many places and focus more on eastern Arabian Sea circulation, change references and figures, will be made and will help to improve and make the ms. more concise.

―――――――――――――――――――――

---

## Author Response (AR1)

**Reply to the reviewers` comments:**

We have substantially modified the ms. and the major changes were:

- "Study area" has become an own chapter and two paragraphs on glacial and Holocene water masses and circulation changes have been added. The differences are later used in order to better understand the reasons for OMZ fluctuations and to understand the difference between the Pleistocene and Holocene conditions.

- A few points about the N cycle were added.

- Model results are presented in the supplement and the methods on model design has been substantially enlarged.

Specific replies to the reviewers:

Reviewer 1:

Page 11, lines 239-247: The available data allowing a comparison of Mg/Ca and alkenone SST are discussed and there is no discernable trend towards a bias to higher or lower SST either of the two methods. As both methods are calibrated by annual average SST we have used both.

Page 15, lines 332-335: The detailed discussion on the shift of the Findlater Jet has helped to better explain the difference between the glacial and Holocene SST patterns. The shift can also explain the prevalence of Oman upwelling while Somali upwelling was possibly shut off. It has moreover, helped to explain the lack of a correlation between Holocene moisture transport, often regarded as monsoon strength, and the SST/productivity pattern.

Differences in circulation are discussed in detail (see above).

Model description has been added including supplementary Figures which show the model performance (see above).

Minor comments

Suggested changes were done, Figures and their labels checked, Figures, Tables and References changed.

The TOC MAR data compiled by Cartapanis et al. show that the pattern in Figure 6g can be found all over the basin. We are discussing the various possible reasons for these changes and conclude that the OMZ intensification in the late mid- and late Holocene are responsible for the observed pattern which deviates from the global TOC MAR pattern.

Reviewer 2:

Abstract

Timings have been checked and corrected throughout the ms.

Introduction

Sea level and atmospheric circulation mentioned as drivers of changes in nutrient inventories.

Materials and Methods

Two paragraphs on water masses have been added and the enlarged "Study Area" is a separate chapter. Ages of Holocene boundaries and reference are given.

Details on the hiatus identification and more details on the age model are given.

Three comparisons of Mg/Ca and alkenone temperatures are made (see above).

Results and Discussion

Ages and precise and consistent terminologies are used.

The discussion of the situation in the eastern Arabian Sea is now more elaborate. The differences between stadials and the normal glacial conditions have been elucidated. Factors influencing d15N (diagenesis, different N sources) have been discussed. The discussion of the impact of ventilation is now more detailed.

Similarities and differences to the GISP record are indicated.

Minor comments

References, figures, tables and captions were checked and there are hopefully no more mistakes.

A presentation of the deep circulation of the Arabian Sea is beyond the scope of this paper and would be highly speculative.

Results of Anand et al. 2008 are discussed but we did not find data and did not receive them from the author.

Reviewer 3

Reviewer 3 questioned whether the binning of data into 1 ka intervals is useful and also points out that some of the records have few age tie points while other records are very highly resolved. Due to these differences in resolution we decided to bin the data in 1 ka intervals. The idea was to use this procedure to eliminate some of the local variations and identify the main trends. However, we selected the highly resolved records and now show them in the supplement to strengthen our points. The changes made in the revised version now makes clearer why it is worthwhile to look at the Holocene and part of the last glacial. This comparison shows the main driving mechanisms of OMZ formation in the Arabian Sea and highlights the unique Holocene circulation pattern.

The introduction to the N cycle and the use of 15N was slightly enlarged.

Sample collection

Line 183: Data on core SL 163 and MC680 will be made available on Pangaea – data repository. They are a small contribution to the d15N records but one of only two SST records available for the Oman upwelling region.

Line 215: The age models of the two cores are now described in more detail.

The d15n data are from Banakar et al. (2005) and the SST curve was from Banakar et al. (2010) made available by V. Banakar.

The normalization procedure was carried out in order to be able to look at changes in d15N sources and eliminate the local biases. In the revised version we added a few sentences on the possible biases. In the supplement we have indicated the average value used for normalization for each core. Averages were in a similar range even when cores did not cover the entire 25 ka but when the main shift in d15N values from the glacial to Holocene conditions was covered. In some cases the records were indeed too short to use the average of the particular core. In this case the average was derived from near-by cores when the d15N values in the available time periods were in the same range. All values used for normalization are shown in the supplementary Tables S1.

A paragraph on water mass structure was added (see above, reply to reviewer 2).

Fig. 4: the time slice was chosen as data availability is best for 17-18 ka and this period is neither a stadial nor an interstadial.

TOC MAR are now discussed in more detail: the observed trend can be found across the entire Arabian Sea and its deviation from the global pattern is explained with increased burial efficiency and OMZ intensification (see review No. 4).

Reviewer 4:

The discussion of water mass structures, glacial and Holocene circulation has now become an important part of the ms.. Seven of ten highly resolved records are shown in the supplement to highlight the millennial scale changes.

Reorganisation of the ms. was described above (new section study area; more on d15N)

Terminology was checked and corrected throughout the ms.

The two southern "west" cores have troubled us for some time during ms. preparation and as the SST records are not very different from the western non-upwelling records off Oman we binned them together with the former. We checked and decided to leave it this way as they are similar and a separation would not add more information. Unfortunately, there are no d15N data from the two southern locations.

Lines 445-460: TOC MAR of Cartapanis et al. 2016 were checked and the same trend was found in all areas of the Arabian Sea with a very small trend in the west and east and a strong trend in the near coastal areas. We have therefore retained this part.

Lines 461-473: Model results and a model data comparison are now presented in the supplement. More information about the model are also presented. As in many earth system models the OMZ in the Arabian Sea is not strong enough.

Discussion and Conclusion

The major findings are now better presented, i.e. the major change in ventilation at the end of the glacial from an unstable to a more stable OMZ and the changes within the Holocene which are evidently not related to productivity changes but mainly to changes in the age of the OMZ water mass. This is supported by data and model results.

**Manuscript with changes marked.**

**Figures changed according to reviewer 1 (no major changes)**

**Table corrected**

**Supplement has 5 new Figures: Figure 3-5 on model performance, Figure 6 and 7 show individual SST and $\delta^{15}$N records of high resolution.**

[revised manuscript text omitted]

**Seite 34: [5] Gelöscht**          **Birgit Gaye**          **17.11.2017 11:54:00**

values in slope sediments were by 2-5 ‰ lower than Holocene $\delta^{15}$N and indicate that denitrification was significantly reduced or absent in the glacial Arabian Sea and that the OMZ was much less intense.